



# Effects of AIR pollution on cardiopuLmonary disEaSe in urban and peri-urban reSidents in Beijing: protocol for the AIRLESS study

Yiqun Han[1,2], Wu Chen[1], Lia Chatzidiakou[3], Li Yan[2], Hanbin Zhang[2], Queenie Chan[2,4], Ben Barratt[2], Rod Jones[3], Jing Liu[5], Yangfeng Wu[6,7], Meiping Zhao[8], Junfeng Zhang[9], Frank J. Kelly[2*], Tong Zhu[1*]
and the AIRLESS team

[1]BIC-ESAT and SKL-ESPC, College of Environmental Sciences and Engineering, Peking University, Beijing 100871, China
[2]Environmental Research Group, MRC Centre for Environment and Health, King's College London, London, UK.
[3]Centre for Atmospheric Science, Department of Chemistry, University of Cambridge, UK
[4]Department of Epidemiology and Biostatistics, MRC Centre for Environment and Health, Imperial College London, London, UK;
[5]Department of Epidemiology, Beijing Anzhen Hospital, Capital Medical University, Beijing Institute of Heart, Lung and Blood Vessel Diseases, Beijing, China;
[6]Peking University Clinical Research Institute, Beijing, China;
[7]The George Institute for Global Health at Peking University Health Science Center, Beijing, China;
[8]College of Chemistry, Peking University, Beijing, China;
[9]Duke Kunshan University, Nanjing, China.

*Correspondence to: Tong Zhu (tzhu@pku.edu.cn), Frank Kelly(frank.kelly@kcl.ac.uk)

**Abstract.** Beijing, as a representative megacity in China, is experiencing some of the most severe air pollution in the world, and its fast urbanization has led to a substantial urban and peri-urban disparities in both health status and air quality. Uncertainties remain regarding the possible causal links between individual air pollutants and health outcomes, with spatial comparative investigations of these links lacking, particularly in developing megacities. In light of this challenge, Effects of AIR pollution on cardiopuLmonary disEaSe in urban and peri-urban reSidents in Beijing (AIRLESS) was initiated with the aim of addressing the complex issue of multipollutant exposures on cardiopulmonary outcomes. The two panel studies deployed included urban and peri-urban elderly Beijing residents recruited from two established cohorts. The project was strengthened further through the measurement of an extensive range of ambient and indoor pollutants during two intensive monitoring campaigns undertaken simultaneously with the collection of health data from the panel participants. This paper presents the novel elements and methodology deployed in the AIRLESS project that addressed gaps in current understanding: namely 1) contrast of the exposure to air pollution in peri-urban and urban areas in a developing megacity, 2) a nested case (hypertensive) – control (healthy) study design to identify potential susceptible population, 3) the detailed assessments of personal exposure to air pollution in diverse indoor and outdoor environments using miniaturised portable platforms; 4) detailed assessment of the chemical composition of particulate matter; and 5) a rich collection of biological markers to understand the underlying mechanisms of health responses to air pollution.



# 1 Background

Air pollution has been widely recognised as a major risk factor for human health especially related to cardiopulmonary morbidity and mortality. According to the Global Burden of Diseases (GBD) study, exposure to ambient particulate matter of aerodynamic diameter ≤2.5 μm ($PM_{2.5}$) contributed to 4.09 million premature deaths worldwide annually, with China suffering the greatest health burden (1.08 million attributed deaths) (WHO, 2017). Chinese megacities, including Beijing, and their surrounding areas, have high population densities and some of the highest air pollution concentrations in the world

(Kelly and Zhu, 2016;Parrish and Zhu, 2009), with $PM_{2.5}$ concentrations regularly exceeding World Health Organization air quality guidelines (Liu et al., 2016a). However, the burden estimates in China were based almost entirely on epidemiological studies undertaken in Europe and North America where concentrations and mixtures of air pollution in urban settings are likely to differ considerably between Western and Chinese cities. It is unclear if health risks relating to poor air quality can simply be transcribed from one setting to the other (Burnett et al., 2014).

The rapid urbanization process especially in some Chinese megacities such as Beijing has resulted in a substantial urban and peri-urban disparities. This is reflected not only in health status due to the differences in social economics, and health services (Li et al., 2016), but also from the spatial contrast in air pollution in the greater Beijing area (Zhao et al., 2009;Wu et al., 2018;Xu et al., 2011). These contrasts in air pollution are partly related to the variation in energy use – e.g. in winter urban areas are dominated by centralized gas heating system, while traditional biomass and coal stoves remain the key

emission source for heating and cooking in peri-urban areas (Liao et al., 2017;Liu et al., 2016b). These contrasts in urban and peri-urban locations due to rapid urbanization provides an opportunity to investigate the difference of local air pollution and its subsequent health impacts on local residents. Such comparative investigations are however largely lacking to date, especially in a rapidly developing country such as China.

A consortium of UK and Chinese researchers developed the project "Effects of AIR pollution on cardiopuLmonary disEaSe

in urban and peri-urban reSidents in Beijing (AIRLESS)" which is nested within the Air Pollution and Human Health in a Chinese megacity Research Programme (APHH, http://aphh.org.uk/). APHH includes four other research themes: (1) Sources and emissions of urban atmospheric pollution, (2) Processes affecting urban atmospheric pollution (Brook et al.) Air pollution and health and (4) Interventions and solutions. Based within two existing cohorts in urban and peri-urban areas of greater Beijing, AIRLESS includes four repeated follow-up clinical measurements in winter and summer, during which

intensive air pollution monitoring campaigns were undertaken simultaneously. Together the APHH programme is compiling a wide range of air pollutants measurements and human health datasets, providing the opportunities to test a variety of hypotheses relating to the adverse cardiopulmonary effect of air pollution.

Beyond the need for urban and peri-urban comparative investigations into exposures, the AIRLESS project is also addressing several cutting-edge challenges and research gaps, as listed below:

a) High blood pressure and air pollution are now ranked as the 3rd and 4th of the leading risk factors of mortality in China (WHO, 2017). Hypertension is epidemic in China as the age-specific prevalence are reported as 13.0%, 36.7%, and 56.5%



among persons aged 20–44, 45–64, and ≥65 years, respectively (Gao et al., 2013). Air pollution, including household air pollution (HAP) from traditional biomass and coal stoves, is also acknowledged to be a strong determinant of blood pressure (Brook et al., 2010), leading to a hypothesis that air pollution directly and indirectly via hypertension is impacting on human health. However, the interactive effects of these two risk factors, i.e. whether hypertensive individuals are susceptible to the adverse effect of air pollution remains unclear (Sacks et al., 2011).

b) Accurate assessment of personal exposure would seem crucial to quantify and interpret the actual health effects of air pollution. However, this metric has been absent in most studies to date, where spatial and temporal variation of air pollutants in different microenvironments have not been taken into account, leading to misclassification of exposure (Jerrett et al., 2005). This uncertainty might even be magnified in peri-urban areas with fewer fixed monitoring sites and reported high levels of HAP, thus biasing the estimation of exposure-response relationship (Steinle et al., 2013). Assessing personal exposure has been challenging because of the expense, difficulties in measuring time-resolved multiple pollutants with a high time resolution, and quantifying exposure-activity logs. However, the rapid advancement in low-cost sensors in recent years has provided potential for a great improvement in this aspect of exposure science.

c) Humans are exposed to a complex mixture of gaseous and particulate pollutants emitted from a range of sources and/or arising from different chemical reactions. Although epidemiological studies worldwide have reported associations between reduced cardiorespiratory health and increased concentrations of air pollutants, such as PM (Brunekreef et al.), ozone ($O_3$) and nitrogen dioxide ($NO_2$) (Brunekreef and Holgate, 2002;Shah et al., 2013), it is recognised that some pollutant species, especially PM with certain size fractions, chemical compositions or source-related mixtures are likely to be more toxic than others (Kelly and Fussell, 2012). However, research to date has yet to clearly define the differential toxicity of particle compositions, owing to the limited high-quality measurements of pollutant species applied in most epidemiological studies, and often the high correlation between multiple pollutants (Han and Zhu, 2015). A better understanding of the chemical and physical properties of air pollutants should lead to a major enhancement in knowledge of the sources and differential health impacts of air pollutants.

d) Although existing epidemiological evidence strongly supports a causal relationship between $PM_{2.5}$ exposure and cardiopulmonary disease, gaps and uncertainties exist in our understanding of why and how this happens, and what the implications may be for air pollution control (Brook et al., 2010). A number of pathways, such as oxidative stress (Lin et al., 2015;Gong et al., 2014), respiratory and systemic inflammation (Han et al., 2016;Lin et al., 2011), autonomic imbalance and vascular dysfunction (Huang et al., 2012) have been suggested as mediators of the cardiopulmonary effects of air pollution. However, few human studies have undertaken thorough examinations on the wide range of biomarkers and the links within the underlying mechanisms (Rajagopalan and Brook, 2012;Brook et al., 2010;Brook and Rajagopalan, 2009).

**2 Methods/Design**



## 2.1 Aims and objectives

The overall aim of the AIRLESS project is to investigate the associations between exposure to multiple air pollutants and changes in health outcomes with a focus in cardiopulmonary biomarkers in urban and peri-urban residents in Beijing. The objectives are:

- To recruit two panels, each comprising of 120 subjects from existing cohorts in urban and peri-urban Beijing;
- To establish infrastructure in urban and peri-urban sites to measure meteorology, gaseous and particulate pollutant
concentrations, including detailed chemical composition and size-fractional particles using state-of-the-art instrumentation;
- To use novel personal air monitors (PAM) and portable instruments to assess personal and residential exposures to key ambient and household air pollutants;
- To assess cardiopulmonary health in the two panels;
- To examine associations between multiple air pollutant species and key cardiopulmonary biomarkers controlling for confounding variables;
- To quantify and compare the air pollution associated health responses between urban and peri-urban areas, and between hypertensive and healthy participants, across winter and summer seasons.

## 2.2 Design and study population

AIRLESS study design is shown in Figure 1. Basically, AIRLESS is designed as a panel study with repeated clinical measurements in both winter and summer seasons. One panel of urban and another panel of peri-urban subjects were recruited from two existing cohorts. Intensive ambient air pollution monitoring campaigns were launched simultaneously close to the participants residences at two urban and one peri-urban locations in Beijing during winter (7th Nov–21st Dec 2016) and summer field visits (22nd May–21st Jun 2017). Detailed descriptions of ambient air pollution monitoring
campaign and clinic examinations are described in section 2.3 and 2.4.

The two existing cohorts approached to construct the AIRLESS panels are the Chinese Multi-provincial Cohort Study (CMCS) (Liu et al., 2004) from urban Beijing and the International Population Study on Macronutrients and BP (Stamler et al.) (Stamler et al., 2003) in peri-urban Beijing.

CMCS was initiated in 1992 with the inclusion of 30,121 Chinese adults aged 35 to 64 years from 11 provinces in China. It
was established with the aim to explore risk factors that contribute to chronic diseases occurrence and progress, mainly focusing on cardiovascular and pulmonary diseases. One of the CMCS population samples is residing within the communities scattered around Peking University (PKU) Hospital, which locates north west to the 4th ring road of Beijing.

INTERMAP study is an epidemiological investigation aiming to clarify the role of multiple dietary factors in the aetiology of unfavourable blood pressure levels prevailing among most middle-aged and older individuals. The cohort comprised of
4,680 men and women aged 40-59 years from 17 diverse population samples in China, Japan, UK and USA, with one sample



from Pinggu district (formerly Pinggu county), a peri-urban area to the eastern end of Beijing with agriculture as a main sector of the local economy. The subjects from Pinggu all reside in a number of local villages. INTERMAP is one of a few cohorts in peri-urban China with historical records on the pattern of energy use. The breadth and depth of high-quality data in both cohorts provide an excellent complement to the new data and bio-samples collected in AIRLESS.

To re-enrol 120 subjects from CMCS and 120 subjects from INTERMAP new infrastructure was established for the clinical examination of participants at the Peking University Hospital (urban site), and at Village Xibaidian, Pinggu (peri-urban site), which are about 70 km apart (Figure 2).

Based on the latest follow-up records for both cohorts, the available sample size of subjects during AIRLESS recruitment period was 1252 (CMCS) and 177 (Stamler et al.). Screening criteria including age, smoking status, health condition and

residential address etc (Table 1.) were agreed for subject recruitment. After screening, potential subject number who met recruitment criteria were 1252 and 88 at the urban and peri-urban site, respectively. As the number of eligible subjects of INTERMAP was insufficient due to the high prevalence of smoking in the existing cohort, we recruited a further 90 non-smokers subjects from the surrounding villages based on the same criteria. In total, the sample size of eligible subjects from peri-urban site was then 178. Potential subjects were randomly contacted through telephone call, or face-to-face meetings to

discuss the project and to make appointments for clinical examinations during the two intensive campaign periods until we hit the target number of 120 participants at each site. Final recruitment figures were, 123 and 127 subjects at the urban and peri-urban clinics respectively. A subgroup of 39 urban and 33 peri-urban subjects were further selected for a pilot residential exposure monitoring as described in section 2.3.3. The detailed screening steps are shown in Figure 3.

Upon recruitment written informed consent was obtained from all subjects prior to study commencement. Following this,

subjects were informed about the study protocol and completed baseline questionnaires (see below). The study protocol was approved by the Institutional Review Board of the Peking University Health Science Centre, China (IRB00001052-16028), and College Research Ethics Committee of King's College London, UK (HR-16/17-3901).

Upon enrolment, the following information was collected through a baseline questionnaire:

- Demographic information (e.g., gender, age, education, income, etc.)

- Current energy use patterns (e.g., types of fuels and stoves, frequency of cooking and heating stove use), and changes in household energy use

- Building characteristics

- Active and second-hand smoking history

- Dietary habit (e.g., consumption of alcohol, coffee/tea, sugar beverage drinking, fried food, vegetables, etc.)

- Sleeping Condition

- Daily activity patterns (transportation, exercise, and potential exposure sources)

- Major health conditions, events, and diagnoses on non-cardiovascular outcomes since the original enrolment

- Regular medication/supplement usage



## 2.3 Measurements of air pollution

### 2.3.1 Ambient exposure

A comprehensive dataset of ambient pollution metrics was collected in both seasons under the monitoring campaigns performed as part of Theme I, II and III (AIRLESS) of APHH research programme. Urban measurements were performed at two existing air quality monitoring stations with historical air pollution data, and peri-urban measurements were obtained from a newly established monitoring site adjacent to the clinic in Pinggu District. The urban and peri-urban clinics were both less than 500 metres away from the nearby monitoring station, and most subjects' residential addresses were in close proximity. The details of the three fixed stations are as followed:

**Urban site PKU**: One of the urban monitoring sites is on the roof of a six-floor building on the PKU campus, namely the Peking University Urban Atmosphere Environment Monitoring Station (PKUERS) (Wang et al., 2018a), which is located at 500 meters north to the 4th ring road (GPS coordinates: 39.990, 116.313).

**Urban site IAP:** The second urban site is located at Institute of Atmospheric Physics (IAP) (Sun et al., 2012), 11 km southeast from PKU site. A 325 metres tall meteorological tower provided the opportunity for vertical measurements of air pollution.

**Peri-urban site:** At peri-urban Beijing, new infrastructure was established for the intensive monitoring campaign at Xibaidian Village, Pinggu District, which is about 75 km northeast to the PKU site (GPS coordinates: 40.167, 117.047). Instruments were deployed on the roof of a one-story building in the far north end of the village.

The same core instruments were deployed at each of the three sites (Table 2) with slight differences for certain pollutants between the sites. A rich set of pollutants including physical and chemical compositions of PM were generated during the monitoring campaign. Meteorological variables including temperature, relative humidity (Collaborators et al.), barometric pressure, and wind speed and direction, and gaseous pollutants (CO, NOx, SO$_2$, and O$_3$) were measured with fast-response online analysers on each site and at six different heights (8, 32, 102, 160, 260, and 320 m) of the IAP tower. To assess the physical and chemical properties of PM, a range of measurements was made to determine its size distribution and chemical composition including black carbon (BC). Daily samples of PM$_{2.5}$ (08:00 AM - 07:30 AM) were collected on Teflon and quartz filters by medium and high-volume samplers during the monitoring campaign periods. The levels of elemental carbon (Jerrett et al.), organic carbon (OC), SO$_4^{2-}$, NO$_3^-$, NH$_4^+$, Na$^+$, K$^+$, Mg$^{2+}$, Ca$^{2+}$, F$^-$, Cl$^-$, water-soluble organic compounds, and polycyclic aromatic hydrocarbons (PAHs) were stored and will be analysed in the laboratory. Details of the instrumentations were elaborated in the APHH programme overview (Shi et al., 2018).

### 2.3.2 Personal exposure

A key methodological strength of the AIRLESS project is the assessment of personal exposure to air pollution at a high spatial and temporal resolution. Taking advantage of recent advancements in sensor technology and computational techniques, a novel highly portable monitor (400 grams) was developed at the University of Cambridge (Figure 4) and has



been successfully applied in a panel of subjects in UK for the adverse effect of personal exposure on chronic obstructive pulmonary disease (COPD) exacerbations (Moore et al., 2016). The PAM operates autonomously, continuously and is almost completely silent. It incorporates multiple low-cost sensors of physical and chemical parameters, as listed in Table 3. The PAM has a battery life of 24 hours, and can be charged on a designated base-station. Measurements are recorded at 1-

min time resolution and stored internally on a secure digital card. The data is then transmitted automatically to a secure server when the PAM is returned to the base-station for daily charging.

Thirty PAM devices were deployed at both the urban and peri-urban clinic sites, which enabled the recruitment of 30 subjects each week. The PAM was deployed in an easy-to-use carry case for protection, and each subject was instructed to carry the PAM for one week of normal daily life. No other interference was required by the subjects, other than to place it in

the base-station each night for charging and data transmission. Subjects were informed that the monitors utilise GPS technology and were reassured that this information would not be accessed in real time, but only used at the end of the study to analyse overall spatial and temporal relationships of fully anonymised data.

The collection of timestamped geo-coordinated measurements together with background noise and accelerometer readings enables the classification of time-activity-location events with an automated algorithm. The algorithm is a progressive

composite tool that employs a spatiotemporal clustering, rule-based models and machine learning techniques. This enables the integration of inhalation rates in the estimation of activity-weighted exposure at the individual level, often used as a proxy for "dose" and will be extremely useful for future epidemiological investigations.

### 2.3.3 Residential exposure

Residential exposures of sub-groups from both AIRLESS panels were measured during both the winter and summer

campaigns. At the urban site, measurements were conducted in the homes of subjects (N = 39) who live within 100 m to the nearest main road. At the peri-urban site, a sub-group of subjects (N = 33) were selected to be representative of AIRLESS peri-urban panel in regard to the cooking and heating methods during winter. Residential exposure was measured only with resident's permission for home access and monitoring.

Two portable real-time monitors, namely, MA300/350 multi-wavelength aethalometer (Aethlabs, USA) and MicroPEM v3.2

(RTI International, USA) were deployed for residential exposure measurements of black carbon and $PM_{2.5}$. Instruments were co-located with reference monitors in Beijing before and after the fieldwork. Operation of the instruments followed strict QA/QC method to ensure data quality. Monitoring instruments were installed in the room where the subject spends most of his/her time, with a consideration for noise tolerance of the residents.

Residential monitoring was designed to be a subset of the one-week personal exposure assessment. The monitoring period

for each subject's home was between three and four days, either between the enrolment and the first clinical examination, or between the first and the second clinical examinations. With such arrangements, home visits happened only on the day of clinical visits so that the disturbance to participant's normal life could be minimized.



### 2.4 Clinical examination

Clinical examinations were performed in both the winter and summer simultaneously with the intensive air pollution
measurement campaigns. Each subject participated in a 7-day follow-up during both seasons (Figure 5). After enrolling at
the clinic in the morning of DAY 0 (e.g. Monday), each subject was instructed to carry a PAM for the next 7 days. During
this week they returned to the clinic for two repeated clinical examinations at 8:00 am on DAY 3 (i.e. Thursday) and DAY 7
(i.e. Monday). Details of the clinical measurements are described below and are listed in Table 4.

**At Day 0:**

• Each subject was provided with a PAM and instructed on its use, and then carried the PAM with them during their daily
activities and keep in the bedroom during night time to obtain a one-week personal exposure

• Basic anthropometry measurements, such as weight, height, hip and waist circumference were obtained.

**At Day 3 and 7** (90 min in clinic):

• Subjects were asked to complete a follow-up questionnaire at DAY-3 and Day-7 on their energy usage, exposure,
activities, medication use and any sleep disturbance over the past three days.

• Three consecutive measurements of brachial artery blood pressure were taken using digital automatic blood pressure
gauge (Omron HEM 907) for each subject in a sitting position after resting for five minutes.

• Three consecutive measurements of vascular function including central (aortic) blood pressure, arterial stiffness
parameters [augmentation pressure (AP), augmentation index (AI), ejection duration (Shi et al.), and the subendocardial
viability ratio (SEVR)] for each subject in a supine position using a pulse wave analysis system developed by Chinese
Academy of Sciences (Zhang et al., 2012).

• Each subject was provided with a peak flow metre (Williams Medical, UK) and was instructed to perform three
consecutive peak expiratory flow (PEF) measurements themselves every morning during the participation week. Daily
measurements were recorded in a diary card together with self-reported respiratory symptoms.

• 4 liters of breath was collected in an aluminium air-sampling bag at a constant exhaled NO ($FE_{NO}$) was measured with a
chemiluminescence nitrogen oxide analyzer (model 42i; Thermo Scientific) at a constant flow rate of 150 mL/s.

• 1 mL of exhaled breath condensate (EBC) was collected using a Jaeger EcoScreen collector (Erich Jaeger, Friedberg,
Germany), and was used for analysis of pH value and inflammatory cytokines.

• Each subject was provided with a 15 mL polypropylene tube and was instructed to collect the midstream of their first
morning urine sample.

• Before blood sample collection subjects were asked to fast overnight (> 12h). All blood samples (2 ml plasma in
EDTA-coated tube, and 4 ml serum in uncoated glass tube) were collected by nurse before 09:30 AM during the
clinical visits.

Urine samples were stored at –20 ℃ and blood samples at –80 ℃ immediately after the collection or pre-treatment (such as
centrifuge and sub-packing done in 2 hours with samples placed on ice).


Counts of white blood cells (WBCs), neutrophils, monocytes, lymphocytes, red blood cells, and haemoglobin and platelets were measured immediately in local clinic after blood collection. Levels of glucose related parameters [fasting glucose, insulin, homeostatic model assessment of insulin resistance (HOMA-IR)], lipid related parameters [triglyceride (TG), high–density lipoprotein (HDL), low–density lipoprotein (LDL), and total cholesterol (Chol)] and C-reactive Protein (CRP) were

measured one month after the end of each campaign in the Anzhen Hospital in central Beijing. Further laboratory analyses included: 1) multiple cytokines in EBC and the remaining blood samples, including interleukin1–α (IL1–α), IL1–β, IL–2, IL–6, IL–8, tumor necrosis factor–α (TNF–α), and interferon–γ (IFN–γ); 2) concentrations of creatinine, malondialdehyde (MDA) and 8-hydroxydeoxyguanosine (8-OHdG) in urinary samples; 3) DNA repair enzymes in plasma samples; 4) High-throughput metabolomic analysis for both plasma and urine samples via gas chromatography–mass spectrometry (GC/MS)

and liquid chromatography–mass spectrometry (LC/MS); 5) Genome-wide association studies were also planned for the second stage analysis, where genetic profiles and epigenomic data will be measured based on whole blood samples.

### 2.5 Sample size and power calculations

One of the main analysis in this study will be examining the associations between air pollutants and the changes in multiple cardiopulmonary biomarkers. Based on a sample size of 240 subject, we examine the minimum detectable effect of $PM_{2.5}$ on

the 4 key health outcomes namely SBP, DBP, $FE_{NO}$ and WBC, given the means and standard deviations (Collaborators et al.) from previous studies (Dubowsky et al., 2006;Han et al., 2016;Jiang et al., 2014). Table 5 and Figure 6 show the minimum detectable effects in cross-sectional and longitudinal settings, with varying within-subject correlation coefficients. The results suggest the assumed sample size will provide adequate power to detect the changes in the 4 key cardiopulmonary outcomes that meet what has been found in previous studies (Dvonch et al., 2009;Han et al., 2016;Dubowsky et al., 2006).

For example, assuming a within subject standard deviation of 7.0 mmHg in SBP, for a SD increase in the level of exposure to $PM_{2.5}$, a two-sided F test at a significance level of 0.05 with a 80% statistical power will be able to detect an increase of 1.25 unit in SBP in a cross-sectional setting, and an increase of 0.53 and 0.34 unit in SBP in a longitudinal setting with the within-subject correlation as 0.5 and 0.8 respectively.

### 2.6 Statistical analysis

In the AIRLESS project we aim to: 1) examine the associations between multiple air pollutants and a wide range of cardiopulmonary changes; 2) compare the difference of biological changes in urban and peri-urban settings across seasons; 3) determine if these associations differ in potential susceptible subjects e.g. those with hypertension or other underlying cardiopulmonary disease.

A master database was built to link the data obtained from ambient, residential and personal exposure to air pollutants, health

outcomes and baseline and follow-up questionnaires. Mixed linear effect models accompanied with distributed lag structures will be applied to examine the associations between air pollutants and health outcomes. The model will include a single random intercept for participant and assumed equi-correlation between all observations assigned to each participant.





Multiple variables will be controlled in the model, including age, sex, body mass index (BMI), smoking status, medication usage, history of diseases, and day of week (Shi et al.). Temperature and RH will also be adjusted with non-linear function

with specific parameters determined by the minimum of Akaike information criterion (Collaborators et al.). We will estimate the changes in biomarker concentration associated with each interquartile range increase in pollutant concentrations in the 24 hours before the clinic visit, as well as the previous 1-7 days. To investigate the difference of biological responses to ambient PM2.5 between urban and peri-urban residents and between potentially susceptible subjects and healthy controls, stratified effect will be estimated by adding an interactive term of exposure and categorical variables of tested group in the model. All

statistical analysis will be performed using R Statistical Software (www.r-project.org).

### 3 Preliminary results

In total, we have recruited 251 subjects (urban = 123, peri-urban = 128), and 218 subjects (urban = 102, peri-urban = 116) have completed all the 4 visits at the end of the summer campaign. In total, 936 person-times clinical visits were collected, and the response rate was 83% (102/123) and 91% (116/128) for re-enrolment in summer campaign in urban and peri-urban

site respectively (Table 6).

One of our aims is to examine if subjects with hypertension are more susceptible to air pollution associated health effects, so we considered to balance the number of hypertensive and healthy subjects during recruitment stage. The ratio of hypertensive to healthy subjects was close in the winter campaign (HBP:Health = 124:127), and ended up as unity (HBP:Health = 109:109) in the summer. However, a higher hypertension ratio was recorded at the urban site (HBP:Health =

56:46) compared with the peri-urban site (HBP: Health = 53:63). The gender ratio in the urban site was unity (F:M = 51:51), in the peri-urban site significantly more females completed the clinical visits in  seasons (F:M = 67:49) resulting in higher female participation rates in the study (F:M = 118:100).

### 4 Discussions

AIRLESS focuses on the adverse health effects of air pollution in Beijing, China, a metropolitan city characterized by a high

population density and poor air quality. Owing to rapid urbanization, Beijing also manifests a unique difference in health status (Li et al., 2016) and air pollution settings between urban and peri-urban areas, both in concentrations and species (Zhao et al., 2009;Wu et al., 2018;Xu et al., 2011), which may, in part, be responsible for different health responses of local residents. Therefore, the results from the AIRLESS project will be a valuable record and opportunity to study the contrast of air pollution by location and as a consequence impact on human health within a major city with high concentrations of air

pollution. The arising results may provide evidence to improve governmental effort on air pollution control in China and other developing countries.

China has underwent rapid health transitions in the last three decades, with a marked decline in child mortality and infectious diseases, and with cardiovascular and pulmonary disease emerging as the leading causes of death (Yang et al., 2013). Hypertension is the leading cardiovascular disease (CVD) risk factor in China, where the reported prevalence in adult

population is 23.4% and 23.1% in urban and rural residents respectively (Wang et al., 2018b). Although many studies have considered hypertension in stratified analyses when examining the association between short-term PM exposure and CVD





related outcomes, it remains unclear if hypertension is a significant modification factor (Sacks et al., 2011). Given the nationwide hypertension epidemic and severe air pollution in China, it is of considerable public health importance to know if people with hypertension are more susceptible to the health effects of air pollution. With hypertensive and healthy subjects

participating in the same follow-up visit for exposure monitoring and health examination, AIRLESS has the potential to examine the susceptibility of hypertensive population to the adverse effect of air pollution, and therefore provide evidence for the improvement of disease burden estimation and prevention.

Besides, taking advantage of the two panel studies and simultaneously performed monitoring campaigns, AIRLESS is compiling a rich dataset of air pollutants including detailed size fraction and chemical compositions of PM, and a wide range

of cardiopulmonary health outcomes in personal level. It is very unique in a health study as it requires a strong design, large team and cooperation to integrate intensive monitoring campaign and health examination. This enables the AIRLESS research team to test a variety of hypotheses, e.g. to understand the relationships between ambient, residential and personal exposures, to identify the toxic air pollutant species, to examine the joint health effect of multiple air pollutants, and to understand the critical biological pathways.

AIRLESS is also strengthened through the use of a state-of-the-art personal monitoring to improve the personal air pollution exposure assessment. The illustrative example in this protocol showed a wide range of personal exposure levels in different microenvironments driven by individual activity patterns, and clearly differs from ambient measurements that would be used to assign exposure to the same individual. The difference might be even more prominent in peri-urban area considering the contribution of HAP from traditional biomass and coal stoves. AIRLESS will also integrate inhalation rates later to obtain an

activity-weighted dose, which will be valuable for exposure-response relationship estimation.

The high compliance rate of the subjects with the study protocol highlighted the feasibility of collecting personal exposure data at high spatiotemporal resolution matched with detailed health assessments. The laboratory analysis of health outcomes and the compiling of exposure and health measurements are still in progress. The improved exposure metrics have the potential to shed light on the identification of the critical toxic species in air pollution mixtures, and the underlying

mechanisms of health responses to air pollution. Altogether, the forthcoming outcomes of the AIRLESS project will enhance our understanding of the impact of environmental exposures on human health in a megacity, and reinforce evidence-based policies that in turn may greatly improve the health and quality of life of China's ageing population

**Abbreviations**

| | | |
|---|---|---|
| 355 | 8-OHdG | 8-hydroxydeoxyguanosine |
| | AIC | Akaike information criterion |
| | APHH | Air Pollution and Human Health programme |
| | BC | Black carbon |
| | BMI | Body mass index |
| 360 | Chol | Total cholesterol |





| | | |
|---|---|---|
| | CO | carbon monoxide |
| | COPD | Chronic obstructive pulmonary disease |
| | CMCS | Chinese Multi-provincial Cohort Study |
| | CRP | C-reactive Protein |
| 365 | CVD | Cardiovascular disease |
| | DOW | Day of week |
| | EBC | Exhaled breath condensate |
| | EC | Elemental carbon |
| | ED | Ejection duration |
| 370 | $FE_{NO}$ | Fractional exhaled NO |
| | GBD | Global Burden of Diseases |
| | GC/MS | Gas chromatography–mass spectrometry |
| | HAP | Household air pollution |
| | HOMA-IR | Homeostatic model assessment of insulin resistance |
| 375 | HDL | High–density lipoprotein |
| | IFN–γ | Interferon–γ |
| | IL | Interleukin |
| | IAP | Institute of Atmospheric Physics |
| | INTERMAP | International Population Study on Macronutrients and BP |
| 380 | LC/MS | Liquid chromatography–mass spectrometry |
| | LDL | Low–density lipoprotein |
| | MDA | Malondialdehyde |
| | NO | Nitrogen oxide |
| | $NO_2$ | Nitrogen dioxide |
| 385 | $O_3$ | Ozone |
| | OC | Organic Carbon |
| | PAH | Polycyclic aromatic hydrocarbons |
| | PAM | Personal air monitors |
| | PEF | Peak expiratory flow |
| 390 | PKU | Peking University |
| | $PM_1$ | Particulate matter of aerodynamic diameter ≤1 μm |
| | $PM_{2.5}$ | Particulate matter of aerodynamic diameter ≤2.5 μm |
| | $PM_{10}$ | Particulate matter of aerodynamic diameter ≤10 μm |
| | RH | Relative humidity |



| | |
|---|---|
| 395 | SEVR | Subendocardial viability ratio |

395    SEVR    Subendocardial viability ratio

TG    Triglyceride

TNF–α    tumor necrosis factor–α

WBCs    White blood cells

## 400  5 Declarations

**Ethics approval and consent to participate**

The study protocol was approved by the Institutional Review Board of the Peking University Health Science Centre, China, and College Research Ethics Committee of King's College London, UK. Written informed consent was obtained from all subjects prior to study commencement.

**Consent to publish**

Not applicable.

**Availability of data and materials**

The datasets generated and/or analysed during the current study are not publicly available due to the requirement of project but are available from the corresponding author on reasonable request.

**Competing interests**

All authors have disclosed that there is no actual or potential competing interests regarding the submitted article and the nature of those interests.

**Funding**

This UK-China collaborative project is jointly supported by external funding bodies, namely National Natural Science Foundation of China (NSFC Grant 81571130100) and Natural Environment Research Council of UK [NERC Grant NE/N007018/1 (1st stage of AIRLESS), NE/S006729/1(2nd stage of AIRLESS)].

The funding from both China and UK is jointly used for the study design, laboratory and data analysis, along with manuscript writing. Specifically, The NSFC funding is mainly used for field work, ambient air pollution monitoring and clinical follow-up in China, and NERC funding is mainly used for coordination, personal monitoring development, data collection and analysis.

**Team List**

Yiqun Han, Wu Chen, Lia Chatzidiakou, Li Yan, Hanbin Zhang, Yanwen Wang, Yutong Cai, Anika Krause, Wuxiang Xie, Yunfei Fan, Teng Wang, Xi Chen, Tao Xue, Gaoqiang Xie, Yingruo Li, Pengfei Liang, Aoming Jin, Yidan Zhu, Yan Luo, Xueyu Han, Xinghua Qiu, Queenie Chan, Ben Barratt, Majid Ezzati, Paul Elliott, Rod Jones, Jing Liu, Yangfeng Wu, Meiping Zhao, Junfeng Zhang, Frank J. Kelly, Tong Zhu

**Authors' contributions**

TZ and FK are co-principle investigators of AIRLESS, designed the study, and revised the manuscript. YH participated in the study design, coordinated air pollution monitoring and clinical measurements in Pinggu site, and drafted the manuscript;



WC coordinated the clinical measurement in PKU site; LC, AK and RJ designed the personal monitor PAM and involved in
the monitor deployment; YH, LY, HZ, XC, YC, WX, AJ, YZ and YL are key staff participated in the clinical measurements
in Pinggu site; WC, YW, TX, YF XH and TW are key staff participated in the clinical measurements in PKU site; HZ and
BB participated in the residential air pollution measurement; XQ, MZ and JZ involved in the design of laboratory
biomarkers. JL coordinated the CMCS cohort; YL, XG and QC coordinated the INTERMAP cohort, ME, PE, RJ, JL, MZ,
JZ, and YW are co-investigators of AIRLESS study and revised the manuscript.

All authors read and approved the final version of the manuscript and ensure this is the case.

## Acknowledgements

We are greatly thankful to all the members of AIRLESS team who helped to accomplish the field work in urban and peri-
urban Beijing site. We also like to thank Prof. Roy Harrison and Dr. Zongbo Shi for organizing and coordination of APHH
programme. We also appreciate AIRPOLL and AIRPRO study team for the collected data of ambient pollutants for
reference calibration and further health analysis.

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



**Tables**

630                                **Table 1: Recruitment Criteria**

| **Inclusion** |
| --- |
| ➢  $50 \leq$ age $\leq 75$ years |
| ➢  Non-smoker or those who has quitted smoking longer than 3 years |
| ➢  Hypertensive subjects (clinical diagnosis*) |
|     Healthy subjects (clinical diagnosis*) |
| **Exclusion** |
| ➢  Diagnosed with disease history of any cardiovascular or metabolic diseases, including Hyperlipidemia, malignant tumour, Coronary Heart Disease, Cardiomyopathy, Arrhythmia, Stroke, Hepatitis A/B, Leukaemia, Biliary calculus, Thyroid nodule, sick sinus syndrome or wearing cardiac pacemaker, Hyperthyroidism, Hypothyroidism, Multiple myeloma, Rheumatoid arthritis, Pancreatitis, Reflux esophagitis, Thyroidectomy |

* Clinical diagnosis: systolic blood pressure (SBP) >= 140 mmHg or diastolic blood pressure (DBP) >= 90 mmHg occurred in two repeated measurements


635            **Table 2: The Matrix of Exposure Parameters in AIRLESS study**

| Exposure Index | Parameters | Method/Instrument | Resolution |
|---|---|---|---|
| **Particulate Pollutants** | PM$_{2.5}$ mass concentration | BAM(Pinggu) / TEOM 1400a (PKU) | Hourly |
| | BC mass concentration | MAAP (Pinggu)/AE33 (PKU) | Hourly |
| | Size distribution | SMPS | Hourly |
| | Online EC/OC | Sunset | Hourly |
| | Metal Element | Xact | Hourly |
| | NR-Chemical Composition | ACSM/TOF-ACSM | Hourly |
| | Water Soluble Ions Water Soluble Organic Acid | Dionex ICS-2500/2000 Liquid Chromatogram | Daily |
| | Metal Element | Thermo X series ICP-MS | Daily |
| | PAHs | Agilent GC-MS | Daily |
| **Gaseous Pollutants** | CO | NDIR/Thermo Model 48i | Minute |
| | NOx | Chemiluminescence/ Thermo Model 42i | Minute |
| | SO$_2$ | Fluorescence/Thermo Model 43c | Minute |
| | O$_3$ | UV absorption/ Thermo Model 49i | Minute |
| **Meteorological Parameters** | Temperature, relative humidity, Barometric pressure, wind speed, wind direction | Met One | Minute |



**Table 3: Performance of PAM**

| Parameter | Method | Monitoring Interval |
|---|---|---|
| Spatial coordinates | Global Positioning System (GPS) | 1 min |
| Background noise | Microphone | 100 Hz |
| Physical activity | Tri-axial accelerometer | 100 Hz |
| Temperature ($^o$C) | thermocouple | 1 min |
| Relative Humidity (Collaborators et al.) (%) | Electrical resistive sensor | 1 min |
| PM1, PM2.5, PM10 (μg/m3) | Optical Particle Counter (OPC) | 20 sec |
| CO, NO, NO2 , O3 (ppb) | Electrochemical sensors (Jerrett et al.) | 1 sec |


**Table 4: Measurement Plans for Health Outcomes in AIRLESS study**

| Biological Pathways | Sample/Device | Health Endpoints |
|---|---|---|
| **Blood Pressure and Heart Rate** | Omron HEM 907 | Systolic Pressure/Diastolic Pressure/Heart Rate |
| **Endothelial Function** | Pulse wave analyzer | AP/AP75/AIx/AIx75/ED/SEVR |
| **Respiratory Inflammation** | Peak flow metre | PEF |
| | Exhaled Breath | FE$_{NO}$ |
| | EBC | pH |
| | | Cytokines: e.g. IL1α/IL1β/IL2/IL6/IL8/TNF-α /IFN-γ |
| **Cardiovascular Inflammation** | Serum | CRP |
| | | Cytokines: e.g. IL1α/IL1β/IL2/IL6/IL8/TNF-α /IFN-γ |
| | Plasma | WBC/neutrophil/monocyte/lymphocyte |
| **Metabolic** | Serum | TG/HDL/LDL/cholesterol |
| | | Glucose/Insulin/HOMA-IR |
| | Serum/Urine | Untargeted/targeted Metabolomic signatures |
| **Oxidative Stress** | Urine | MDA/Creatinine |
| | Plasma | DNA repair enzyme |
| **Genetic related pathways** | Blood | Genetic and Epigenomic profiles |




**Table 5: The minimum detectable effects for the 4 cardiopulmonary outcomes in cross-sectional and longitudinal settings**

| Health Outcome | Mean± SD[a] | Power | Cross-Sectional[b] | Longitudinal[c] | | | |
|---|---|---|---|---|---|---|---|
| | | | | $\rho$ =0.5 | $\rho$ =0.6 | $\rho$ =0.7 | $\rho$ =0.8 |
| **SBP** | 115.4 ± 7 | 0.8 | 1.25[d] | 0.53 | 0.47 | 0.41 | 0.34 |
| | | 0.9 | 1.44 | 0.61 | 0.54 | 0.47 | 0.38 |
| **SDP** | 68.2 ± 5.2 | 0.8 | 0.93 | 0.39 | 0.35 | 0.31 | 0.25 |
| | | 0.9 | 1.07 | 0.45 | 0.4 | 0.35 | 0.28 |
| **FE$_{NO}$** | 11.3 ± 6.9 | 0.8 | 1.23 | 0.52 | 0.47 | 0.4 | 0.33 |
| | | 0.9 | 1.42 | 0.6 | 0.54 | 0.46 | 0.38 |
| **WBC** | 6.54 ± 1.69 | 0.8 | 0.3 | 0.13 | 0.11 | 0.1 | 0.08 |
| | | 0.9 | 0.35 | 0.15 | 0.13 | 0.11 | 0.09 |

[a]The mean±sd for each biomarker are referred from previous studies;
[b]Two-sided F test for slope at a significant level of 0.05 using simple linear regression with subject N=240
[c]Two-sided F test for the fixed effect of environmental exposure under longitudinal setting with varying within subject correlation with subject N=240 and 4 repeated observation per subject, at a significant level of 0.05 and statistical power of 0.8 and 0.9.
[d]The minimum detectable effects are reported with SD increase in exposure level.



**Table 6: Conditions of Clinical Visit Completion in AIRLESS**

| Urban Site | Winter | | | Summer | | |
|---|---|---|---|---|---|---|
| | HBP | Health | Subtotal | HBP | Health | Subtotal |
| Male | 32 | 26 | 58 | 30 | 21 | 51 |
| Female | 33 | 32 | 65 | 26 | 25 | 51 |
| Subtotal | 65 | 58 | 123 | 56 | 46 | 102 |
| Peri-urban Site | Winter | | | Summer | | |
| | HBP | Health | Subtotal | HBP | Health | Subtotal |
| Male | 25 | 26 | 51 | 23 | 26 | 49 |
| Female | 34 | 43 | 77 | 30 | 37 | 67 |
| Subtotal | 59 | 69 | 128 | 53 | 63 | 116 |
| Total | Winter | | | Summer | | |
| | HBP | Health | Subtotal | HBP | Health | Subtotal |
| Male | 57 | 52 | 109 | 53 | 47 | 100 |
| Female | 67 | 75 | 142 | 56 | 62 | 118 |
| Subtotal | 124 | 127 | 251 | 109 | 109 | 218 |









**Figures**

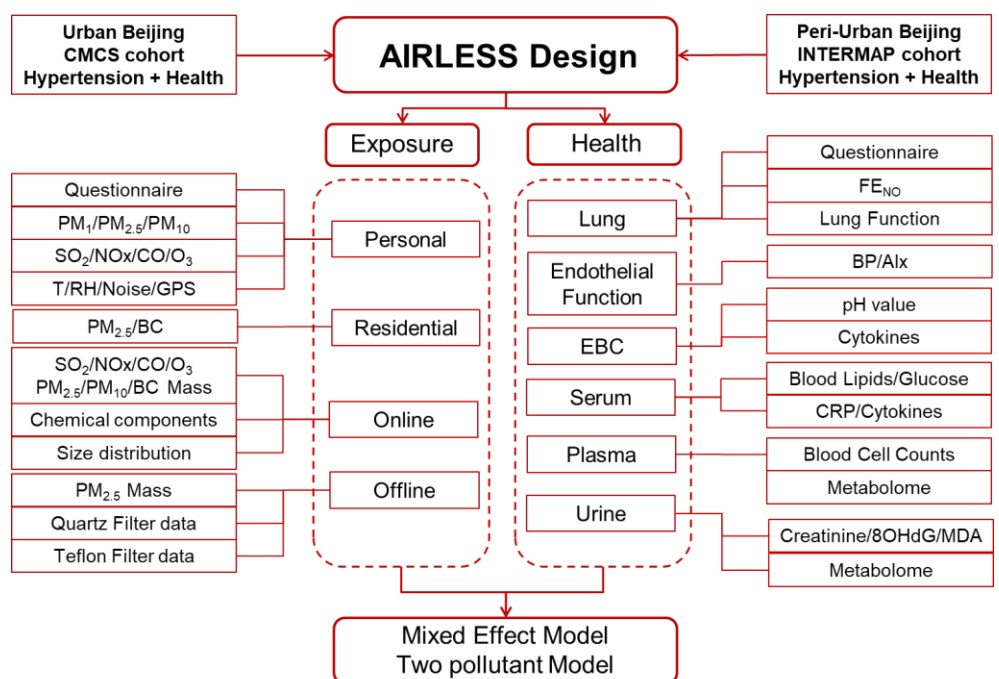

680          **FIG. 1. Design Scheme of AIRLESS**





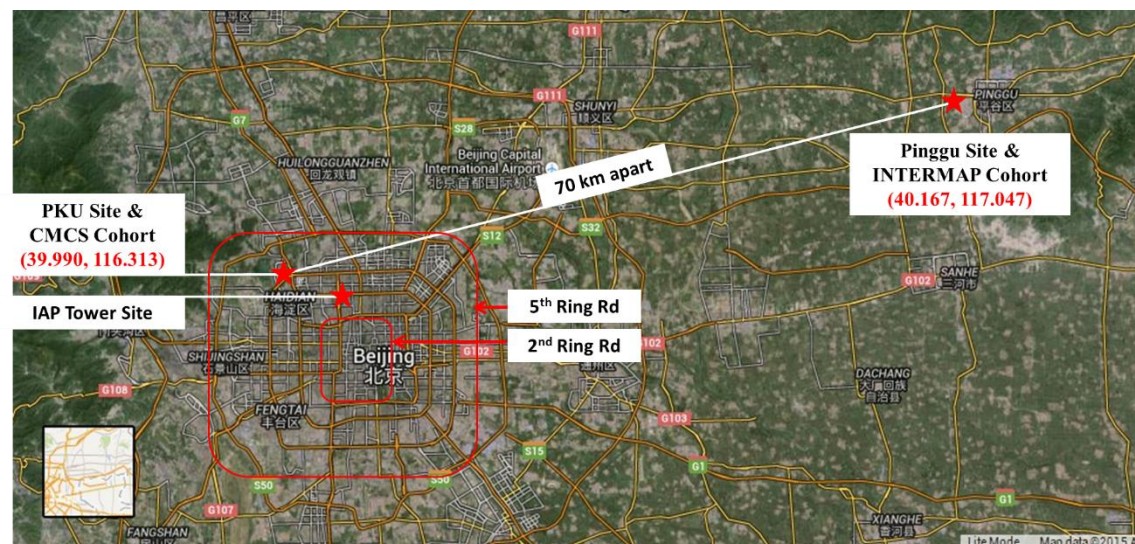

**FIG. 2. Locations of the two cohorts and three monitoring sites in urban and peri-urban Beijing. The figure is based on © Google Maps .**






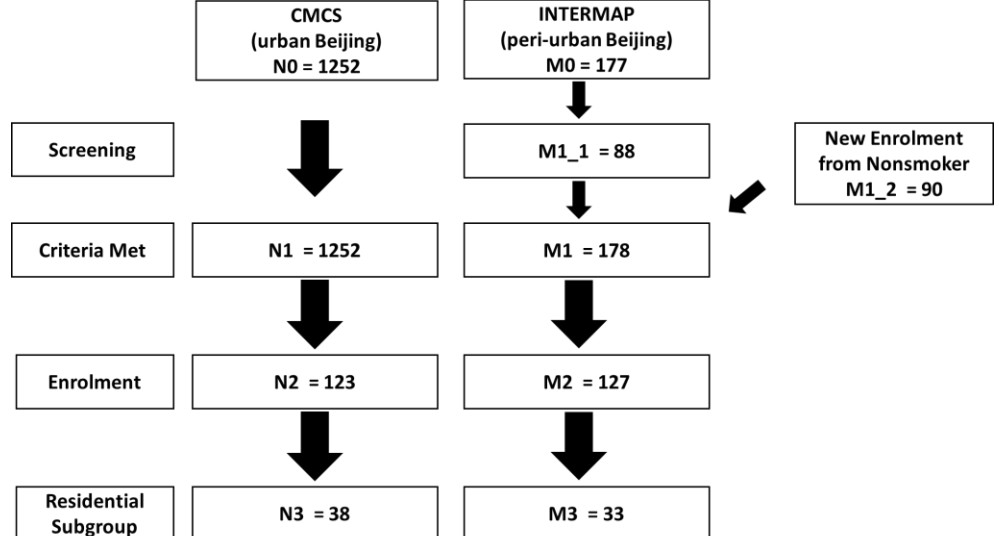

**FIG. 3. Screening steps for Recruitment in AIRLESS. N refers to the sample size of CMCS cohort, M refers to that of INTERMAP cohort; the number after letter N and M refers to the screening layer.**




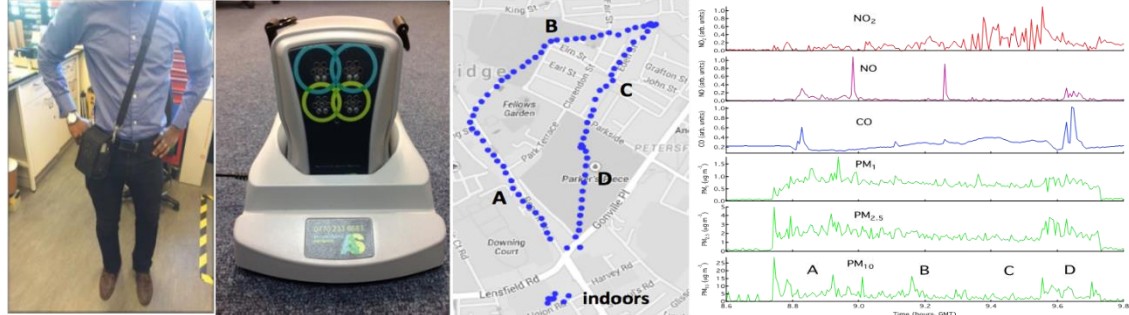

**FIG. 4. Personal Air Monitor (PAM) used in AIRLESS - Model and Applications**





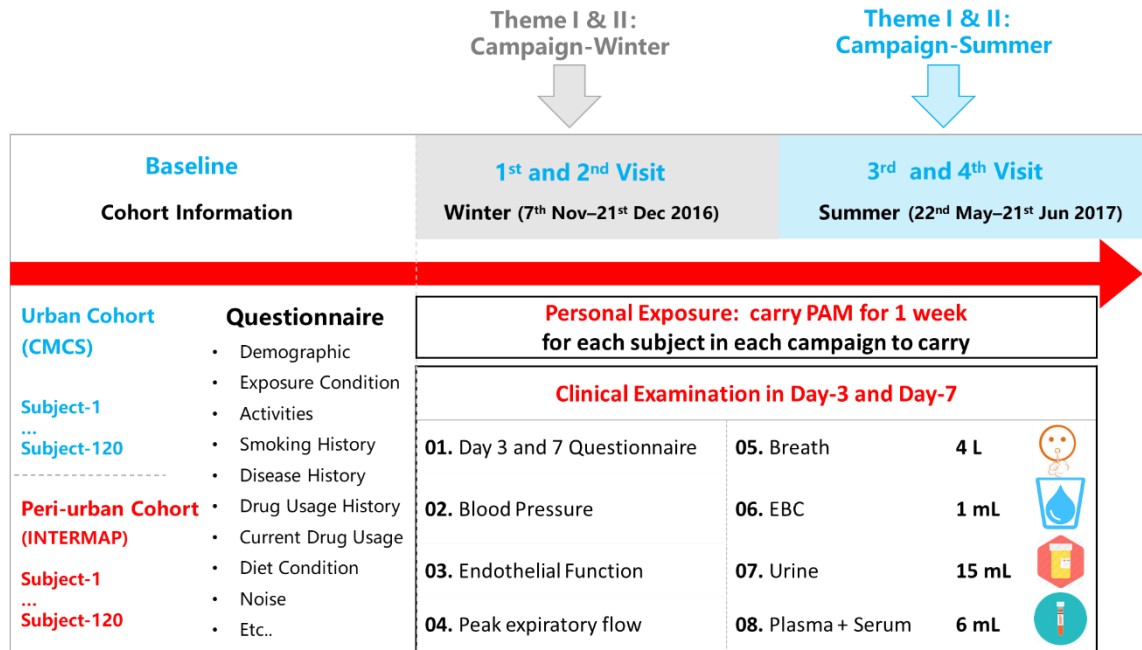

**FIG. 5. Scheme of Clinical Examination of AIRLESS**





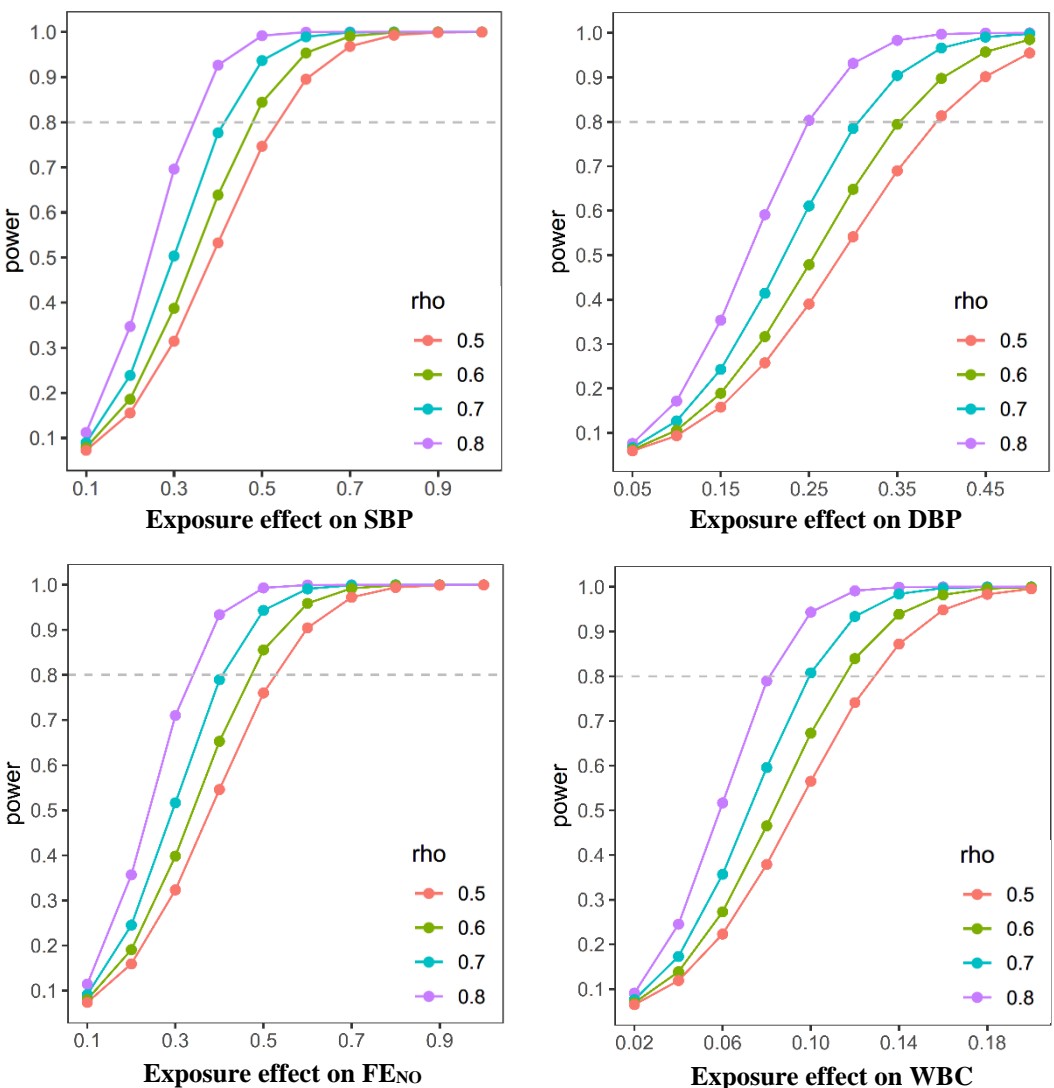

**FIG. 6. Statistical power as functions of the detectable effect and covariance structure for cardiopulmonary outcomes. Power curves are calculated by using a sample size of 240, within-subject correlation coefficients (as denoted by rho) at 0.5 and 0.8, and estimated SDs 7.0, 5.2, 6.9 and 1.69 for outcomes SBP, DBP, FE$_{NO}$ and WBC respectively. Changes in outcomes are reported with SD increase in exposure factor.**