# Peer review of "Effects of AIR pollution on cardiopuLmonary disEaSe in urban and peri-urban reSidents in Beijing: protocol for the AIRLESS study"

_Atmospheric Chemistry and Physics, 2020_

## Referee Comment (RC1) · Anonymous Referee #1 · 14 Jul 2020

This paper presented a methodology/protocol of an epidemiological study during two large air pollution monitoring campaigns (APHH) in both urban and peri-urban areas in Beijing during two seasons (winter and summer). The author elaborated on the study design for both exposure and health measurement. The design of this study is quite complex in terms of the usage of portable monitors for personal exposures, in coordination with the intensive monitoring campaign period, and comprehensive examinations of health outcomes. The protocol shows the strength of combining the panel study with monitoring campaigns which provide the potential to investigate the health effect of detailed chemical compositions and biological mechanisms. It would be useful for other researchers to carry out further studies.

[Figure]

Although it's a protocol paper, it's still necessary to present some preliminary results to show the quality of the data collection and general information of the study. The current result part has summarized the information of participation, and the calculation of sample size, but it would be helpful to add more results to both exposure and health measurements. The following is the main concerns for this paper: 1. For the health part, it's important to know the basic demographic statistics of the participants in both urban and peri-urban sites. E.g. the attended clinical visits, distributions of age, gender, socioeconomic status, baseline exposure status, etc. Is there any significant difference between the two groups? Are you going to compare to the two groups of participants or treat them as two different cohorts? In addition, it would be useful to have some descriptive results related to the measurements of health outcomes.

2. For personal exposure, it's crucial to have some results to validate the performance of PAM with reference instruments. How did you calibrate the instruments, and how well they agree with the reference instruments? What's the measurement range and error? What's the performance of PAM for different microenvironments (i.e. indoor and outdoors)? It's also important to know the completeness of personal exposure monitoring (e.g. how many validate days for the personal dataset, etc), as carrying personal monitors for 7 days is not common in an epidemiological study which would cause a lot burden.

3. A summary of key air pollutants in both urban and rural sites during the health campaign periods in two seasons would lead the readers with a better understanding of the background AP settings, which can be useful to compare with other health studies around the world.

4. The results of comparing personal and ambient exposure with examples from certain participants would be interesting to see the exposure difference, which is one of the main contributions to the uncertainties in the exposure-health relationship.

5. The application of personal measurements is increasing in the epidemiological studies, it's good to add some reviews on the progress in this area to highlight the advantage of this design.

In summary, the paper has shown the uniqueness and detailed methodology of this panels study under the intensive monitoring campaign, however, I suggest the authors restructure the manuscript and add more results for further review and for considerations of ACP publication.

---

## Referee Comment (RC2) · Anonymous Referee #2 · 16 Jul 2020

This is an introduction paper on the protocol of the AIRLESS study conducted in Beijing. The overall study design is rigorous in terms of the methods presented in exposure and health outcome measurements. The comparison of the health effects of air pollution in hypertensive population between urban and suburban area is innovative and interesting. The scope to explore comprehensive range of exposure and health outcome metrics is the strength of the study. Despite of this, there are several places in the manuscript that need to be improved by better clarifications of the methodology and preliminary results. My major concerns are provided as below:

1) There are many air pollutants and health outcome measurements presented in this

study. For air pollutants, it is possible that many species may come from the same sources and therefore, highly correlated. The association, if identified, may not directly reflect the true toxicity of health effect for a pollutant but an alternation of source-related effect. It is important to think more in the paper about how to make use of the comprehensive exposure data and propose novel method that can leverage the combination of multiple pollutants' effect in the health analysis. For the health outcomes, the multiple biomarkers from the same pathways may generate an issue of multiple comparison (or not). As this is a methodology paper, it would be helpful to include a discussion on this issue.

2) The study is unique for being able to use real-time low-cost sensor for personal exposure assessment. However, sensor technology is complex and requires careful calibration both internally within device and externally across the devices and with other standard instruments under various environmental circumstances. The current study only reported the specifications and performances of the PAM monitor, but did not include detailed descriptions on how to ensure the accuracy of the monitors in the real-world measurements. Although personal monitoring may provide better information on micro-environmental exposure for air pollution, it may also produce much larger noise and uncertainties (e.g. systematic or random error) during the measurements which contribute to the health analysis. Such misclassification, if not well-regulated with a valid calibration protocol, may not be less than use of a routine monitoring station for analysis. In summary, it is important for this study to include detailed discussion on QA/QC of the PAM monitors to ensure high quality monitoring data.

3) The results are relatively simple. At least, the demographics of the study population and the exposure and outcome measurement statistics are needed, so that it is good for readers to understand the overall differences of exposure and outcomes between the two study sites. The results will also help support the proposed hypothesis of the study.

4) As mentioned by the authors, one of the major differences in the urban and suburban

sites is the contribution of indoor exposure to the personal exposure and the indoor exposure levels are supposed to be higher. However, only the outdoor monitoring sites include detailed air pollutant species as compared to the personal exposure. Thus, the importance of the contributions of the species to the personal exposure seems to be attenuated. Will it be possible to use the GPS data to split outdoor and indoor exposure in the health analysis, so that the comparisons are relatively fair?

———————————————————

---

## Author Comment (AC1) · 28 Aug 2020

Response to referee #1

Overall: Thank you for your comments and useful suggestions to improve our manuscript. We acknowledged that despite this is an overview paper of AIRLESS project with the focus on methodology, more results are warranted for the readers to understand the studied participant, including their demographic characteristics, and the levels of exposure and health outcomes. Therefore, we have restructured the paper and added more preliminary results accordingly. Please find below the point-to-point response. Detailed table and figures will be finalized and added in the updated

manuscript.

Comment 1: For the health part, it's important to know the basic demographic statistics of the participants in both urban and peri-urban sites. E.g. the attended clinical visits, distributions of age, gender, socioeconomic status, baseline exposure status, etc. Is there any significant difference between the two groups? Are you going to compare to the two groups of participants or treat them as two different cohorts? In addition, it would be useful to have some descriptive results related to the measurements of health outcomes.

Thank you for this suggestion. We have added more analysis to summarize the key characteristics (age, BMI, gender, smoking, annual income, education level), health biomarkers, with an exemplary table as attached. Detailed table and figures will be finalized and added in the updated manuscript. Generally, we treated the residents in urban and peri-urban area of Beijing as two different groups of participants. The characteristics of both ambient air pollution/sources, health conditions, and baseline demographic characteristics are different. Generally, compared with peri-urban participants, urban residents were older, had a lower BMI, and a higher educational and income level at the baseline, as shown in the attached table.

Comment 2: For personal exposure, it's crucial to have some results to validate the performance of PAM with reference instruments. How did you calibrate the instruments, and how well they agree with the reference instruments? What's the measurement range and error? What's the performance of PAM for different microenvironments (i.e. indoor and outdoors)? It's also important to know the completeness of personal exposure monitoring (e.g. how many validate days for the personal dataset, etc), as carrying personal monitors for 7 days is not common in an epidemiological study which would cause a lot burden.

Thank you for this comment. We have added a summary of a previous publication which characterised the performance of the sensors integrated in the PAM thoroughly.

Briefly, the collocation experiment has considered different scenarios (outdoor, indoor, transportaiton)ïijŇand the statistics of measurement range and error were also reported under different microenvironment. The corresponding changes in the method section of the manuscript is as below: All PAMs were calibrated with two outdoor colocation deployments at the PKU urban site after the winter and summer deployments to participants. The performance of the NO2 and PM2.5 sensors was additionally characterised in an indoor microenvironment next to commercial instruments. Overall, the air pollution sensors showed high reproducibility (mean adj R2= 0.93, min–max: 0.80–1.00) and excellent agreement with standard instrumentation. In the winter co-location deployment adj.R2>0.84 for all sensors, while in the summer adj R2>0.71. There were some indications that the EC sensor performance is less reliable at high temperatures (>40°C); however, such extreme thermal conditions were not recorded during the deployment to participants. Further work showed that the error of the PAM was negligible compared with the error introduced when deriving exposure metrics from fixed ambient monitoring stations close to the participants' residential addresses. Hence, novel sensing technologies such as the ones used here are suitable for collecting highly resolved personal exposure measurements in large-scale health studies.

Comment 3: A summary of key air pollutants in both urban and rural sites during the health campaign periods in two seasons would lead the readers with a better understanding of the background AP settings, which can be useful to compare with other health studies around the world.

Thank you for this comment. We have added in the manuscript a whisker box plots illustrate ambient concentrations of CO and PM2.5 measured at the reference monitoring stations and personal concentrations at the urban and peri-urban sites during the winter (Nov-Dec 2016) and summer (May-June 2017) campaigns. Detailed statistics of other pollutants measured concurrently in PAM and fixed monitoring station will be summarized in a supplement table.

Comment 4: Comparison of personal exposure with examples from certain participants, and capture rate of personal exposure data.

We appreciated this comment. An exemplary plot of personal exposure to multiple pollutants of a certain participant (U123) was added in our manuscript, as attached. The plot also includes the ambient concentration of the corresponding pollutant, along with the time-activity (i.e. indoor vs. outdoor) to illustrate the difference between personal and ambient exposure.  

Response to referee #2

Overall: Thank you for taking the time to provide valuable feedback. This paper presented the methodological framework for the collection of detailed medical biomarkers and exposure estimates as part of the AIRLESS project (Beijing, China). The comments are extremely relevant as the reviewer has a clear understanding of current gaps in our understanding of the effects of air pollution on health and the potential of projects like AIRLESS to address such issues. Regarding the reviewer's comment, we have added more tables and figures into the results and discussion session. The manuscript has also been restructured and certain sections has been revised accordingly. Please find below the point-to-point response.

Comment 1a: "The association, if identified, may not directly reflect the true toxicity of health effect for a pollutant but an alternation of source-related effect."

This comment hits the nail on the head. We have re-written parts throughout the manuscript to stress this point. We have modified the background section to stress the four wider research gaps this project aims to address: a) To investigate the interactive effects of air pollution and hypertension. b) To establish more reliable links between air pollution and health effects by reducing exposure misclassification. c) To differentiate source-related health effects of air pollution. d) To investigate the underlying mechanism of air pollution on health. Using novel sensor technologies, we capture personal exposure at high spatial and temporal resolution and break the correlation between traffic related pollutants such as PM2.5 and NO2 often observed at fixed monitoring

sites (due to common emission sources). By reducing the correlation between individual pollutants, we can assign specific health effects to individual pollutants (or to different sources, i.e indoor NO2 has different sources than outdoor NO2 and therefore is a proxy for different pollutants).

Comment 1b: "For the health outcomes, the multiple biomarkers from the same pathways may generate an issue of multiple comparison (or not)."

Thanks for the comment from the reviewer. We reckon the number of biomarkers in the study will increase in the difficulties in explaining the biological mechanisms, as some of the biomarkers may share similar pathways or be regulated in a more complicated biological network (eg. Cytokines). The fast development of omic-related analysis, which could generate thousands of biomarkers, will be helpful but meanwhile add more challenge in understanding the biological mechanism. We have considered this issue for the analysis of multiple biomarkers. Specifically, to investigate the associations between the exposure to air pollutant and the changes in metabolome and transcriptome, false discovery rate (FDR) adjusted p-value will be applied to detect the statistical significance. Pathway enrichment analyses based on the changes in multiple biomarkers will be used to investigate potential mechanisms.

Comment 2: regarding to low-cost sensor technologies, "sensor technology is complex and requires careful calibration both internally within device and externally across the devices and with other standard instruments under various environmental circumstances. The current study only reported the specifications and performances of the PAM monitor, but did not include detailed descriptions on how to ensure the accuracy of the monitors in the real world measurements."

Thanks for the comment from the reviewer. We understand the importance of the validation of the personal monitor (PAM) we used in this study. The performance of the sensors integrated in the PAM has been characterised extensively in a previous publication. To clearly describe the performance of PAM and the collocation experiment,

a paragraph has been added in the manuscript as below: All PAMs were calibrated with two outdoor co-location deployments at the PKU urban site after the winter and summer deployments to participants. The performance of the NO2 and PM2.5 sensors was additionally characterised in an indoor microenvironment next to commercial instruments. Overall, the air pollution sensors showed high reproducibility (mean adj R2= 0.93, min–max: 0.80–1.00) and excellent agreement with standard instrumentation. In the winter co-location deployment adj.R2>0.84 for all sensors, while in the summer adj R2>0.71. There were some indications that the EC sensor performance is less reliable at high temperatures (>40°C); however, such extreme thermal conditions were not recorded during the deployment to participants. Further work showed that the error of the PAM was negligible compared with the error introduced when deriving exposure metrics from fixed ambient monitoring stations close to the participants' residential addresses. Hence, novel sensing technologies such as the ones used here are suitable for collecting highly resolved personal exposure measurements in large-scale health studies.

Comment 3: The results are relatively simple. At least, the demographics of the study population and the exposure and outcome measurement statistics are needed, so that it is good for readers to understand the overall differences of exposure and outcomes between the two study sites. The results will also help support the proposed hypothesis of the study.

Thanks for the comment, apart from the recruitment summary in the manuscript, we have added more analysis to summarize the key characteristics (age, BMI, gender , smoking, annual income, education level), health biomarkers, as well as a boxplot to compare the exposure level between urban and peri-urban subjects in the manuscript, with an exemplary table as attached. Detailed table and figures will be finalized and added in the updated manuscript.

Comment 4: Only the outdoor monitoring sites include detailed air pollutant species as compared to the personal exposure. Thus, the importance of the contributions of the

species to the personal exposure seems to be attenuated. Will it be possible to use the GPS data to split outdoor and indoor exposure in the health analysis, so that the comparisons are relatively fair? Thanks for the comment, this is really an important question to understand the health effect of pollutant species. Currently, personal sensors were not applicable to a wide range of pollutant species, especially considering the performance of measurement at a high time-resolution. The common commercial portable monitors used in most of the epidemiological studies are usually targeted on either particles (PM2.5, BC, etc), or gaseous pollutants (NO2, CO etc). The personal monitor we used was developed ourselves which is unique to include both PM and gaseous sensors in one device. This enable PAM to measure PM in different size fraction, and four species of gaseous pollutants, which will help us to understand the health effect of the most concerned pollutants. We have updated Table 2 in the manuscript to describe the physical and chemical parameters of both ambient and personal exposure measurement. Apart from that, we have also considered the suggestion the reviewer of splitting indoor/outdoor exposure to give a more accurate exposure assessment. A automated model was developed to classify time-activity-location patterns based on parameters collected with the PAMs (GPS, background noise, acceleration)2, which has been described in our newly published paper. The classifications include core location categories: "home", "work", "other indoor static", "other outdoor static", " travel", as well as activities "cooking", "sleeping" and modes of transport ("walk", "cycle", "motorbike", "car/bus", "train/tube"). We have now created a new subsection to highlight this methodological element of the project.

References: 1 Chatzidiakou, L., Krause, A., Popoola, O. A., Di Antonio, A., Kellaway, M., Han, Y., ... & Fan, Y. (2019). Characterising low-cost sensors in highly portable platforms to quantify personal exposure in diverse environments. Atmospheric measurement techniques, 12(8), 4643. 2. Chatzidiakou, L., Krause, A., Popoola, O. A., Di Antonio, A., Kellaway, M., Han, Y., ... & Fan, Y. (2019). Characterising low-cost sensors in highly portable platforms to quantify personal exposure in diverse environments. Atmospheric measurement techniques, 12(8), 4643.

Please also note the supplement to this comment:
https://acp.copernicus.org/preprints/acp-2020-208/acp-2020-208-AC1-supplement.pdf

---

## Author Response (AR1)

**Response to referee #1**

This paper presented a methodology/protocol of an epidemiological study during two large air pollution monitoring campaigns (APHH) in both urban and peri-urban areas in Beijing during two seasons (winter and summer). The author elaborated on the study design for both exposure and health measurement. The design of this study is quite complex in

- 5 terms of the usage of portable monitors for personal exposures, in coordination with the intensive monitoring campaign period, and comprehensive examinations of health outcomes. The protocol shows the strength of combining the panel study with monitoring campaigns which provide the potential to investigate the health effect of detailed chemical compositions and biological mechanisms. It would be useful for other researchers to carry out further studies. Although it's a protocol paper, it's still necessary to present some preliminary results to show the quality of the data collection
- 10 and general information of the study. The current result part has summarized the information of participation, and the calculation of sample size, but it would be helpful to add more results to both exposure and health measurements. Overall: Thank you for your comments and useful suggestions to improve our manuscript. We acknowledged that despite this is an overview paper of AIRLESS project with the focus on methodology, more results are warranted for the readers to understand the studied participant, including their demographic characteristics, and the levels of exposure and health outcomes.
- Therefore, we have restructured the paper and added more preliminary results accordingly. The word has also been revised for a clear expression throughout the manuscript. Please find below the point-to-point response.
   Comment 1: For the health part, it's important to know the basic demographic statistics of the participants in both urban and peri-urban sites. E.g. the attended clinical visits, distributions of age, gender, socioeconomic status, baseline
- exposure status, etc. Is there any significant difference between the two groups? Are you going to compare to the two 20 groups of participants or treat them as two different cohorts? In addition, it would be useful to have some descriptive results related to the measurements of health outcomes.

Thank you for this suggestion. We have added a table (Table 4 in revised version) to summarize the demographic characteristics of urban and peri-urban participants. Basic health outcomes (BMI, WHR, hypertension status) were also included in Table 4. For the other detailed biomarkers, we plan to included in the following papers. Regarding to the exposure

25 in the two sites, we added two figures (Figure 6 and 9 in revised version) to show the ambient and personal exposures during the campaigns in both sites.

We also revised the manuscript regarding to the results of these analysis, which is described in subsection 3.1 "Demographics characteristics of urban and peri-urban participants", 3.2 "Ambient concentration of PM2.5 during study periods", and 3.5 "Seasonal and spatial pattern of the difference between personal and ambient exposure".

30 Generally, although we managed to keep the recruited urban and peri-urban participants in this study balanced in gender and hypertension ratio, we did observe significant differences in many ways (e.g. social-economic status, ambient air pollution

levels, health conditions, etc). Given the similarities and difference between urban and peri-urban sites, we would examine the health effect while using sites as a modifier to see the difference, and will also investigate the reasons behind the difference.

- Comment 2: For personal exposure, it's crucial to have some results to validate the performance of PAM with reference instruments. How did you calibrate the instruments, and how well they agree with the reference instruments? What's the measurement range and error? What's the performance of PAM for different microenvironments (i.e. indoor and outdoors)? It's also important to know the completeness of personal exposure monitoring (e.g. how many validate days for the personal dataset, etc), as carrying personal monitors for 7 days is not common in an epidemiological study which would cause a lot burden.
- 40 Thank you for this comment. We understand the importance of the validation of personal monitor used in this study. Our previous paper (Chatzidiakou et al., 2019) has elaborated the corresponding information in detail, including the measurement performance (range and error), and the validation also considered under different conditions and microenvironment. We added a paragraph in the subsection of method (2.5 Personal exposure in revised manuscript) to summarize the key results from that paper, detailed as below:
- 45 "The characterisation of the performance of the air quality sensors integrated in the PAM is presented in a previous publication (Chatzidiakou et al., 2019). Briefly, all PAMs were calibrated in two outdoor co-location deployments at the urban PKU site next to reference instrumentation for one month after the winter and summer deployments to participants. The performance of the NO2 and PM2.5 sensors was additionally characterised in an indoor microenvironment next to commercial instruments. Overall, the air pollution sensors showed high reproducibility (mean R2= 0.93, min–max: 0.80–1.00) and excellent agreement
- 50 with standard instrumentation (R2>0.84 for all sensors in winter, while R2>0.71 in summer). Further work (Chatzidiakou et al., 2020) showed that the error of the PAM was negligible compared with the error introduced when deriving exposure metrics from fixed ambient monitoring stations close to the participants' residential addresses. Hence, novel sensing technologies such as the ones used here are suitable for collecting highly resolved personal exposure measurements in large-scale health studies." Regarding to the completion of the personal measurement, all participants have completed 3548 personal-days measurements
- 55 (~3.5 million observations in 1-min time resolution). The participants showed high compliance with the protocol, with a mean capture rate of personal data of > 86%.

We added a new sub-section (3.3 Completion of personal exposure during study periods) and a figure (Figure 7 in revised manuscript) to illustrate the results.

Comment 3: A summary of key air pollutants in both urban and rural sites during the health campaign periods in two seasons would lead the readers with a better understanding of the background AP settings, which can be useful to compare with other health studies around the world.

Thank you for this comment, we agreed with you that it's important to have a general idea of the pollution level for this study. We added a figure (Figure 7 in revised manuscript) using PM2.5 as an example to show the ambient level in both urban and rural site during the monitoring campaign.

65 A paragraph (subsection 3.2 Ambient concentration of PM2.5 during study periods) was also added in the manuscript for description.

"Figure 6 shows the ambient  $PM_{2.5}$  concentration during AIRLESS campaigns in winter and summer with a comparison between sites. A clear seasonal trend with a large variation of ambient  $PM_{2.5}$  concentration was observed. Specifically, during winter, the mean (SD) daily concentrations were 132.3 (104.8) µg m-3 and 87.4 (79.0) µg m-3 in the peri-urban and urban site

- 70 respectively, which were significantly higher than the corresponding concentrations in summer as 35.2 (15.0) and 45.1 (20.8) μg m-3. The degraded ambient air quality and several high PM2.5 pollution events in winter were due to the greater stagnation and weak southerly circulation suggested by synoptic-scale meteorological analysis (Shi et al., 2019). The number of days with concentrations exceeding the Chinese standard of 75 μg m-3 was 29 and 19 during winter in peri-urban and urban sites respectively. The PM2.5 concentration in the urban area was constantly lower than the peri-urban site during winter, but the 75 trend was opposite in summer."
  - Comment 4: Comparison of personal exposure with examples from certain participants, and capture rate of personal exposure data.

We appreciated this comment. An example plot of personal exposure to multiple pollutants of a selected case participant (U123) was added to our manuscript (Figure 8 in revised manuscript). The plot also includes the ambient concentration of the

80 corresponding pollutant, along with the time-activity (i.e. indoor vs. outdoor) to illustrate the difference between personal and ambient exposure.

Accordingly, a paragraph (subsection 3.4 An illustrative example of exposure misclassification) was added in results section, as followed:

- "A representative participant was selected to illustrate the concept of exposure misclassification in Figure 8. Personal exposure measurements of example participant U123 participating during the winter campaign are compared with data from the closest monitoring station to the participant's home location (< 5 km). The time-activity model (Section 2.6) determined when the participant was located at home (top row). The personal CO, NO, NO2 and PM2.5 concentrations regularly exceeded the outdoor levels, indicating that strong indoor emission sources (such as a gas stove) operated at regular times. The sources caused personal exposures up to 10 times higher than the ambient pollution levels. When no emission sources were active, the indoor
- 90 CO and NO concentrations approached the outdoor concentrations, whereas the NO2, O3 and PM2.5 were much lower than the outdoor concentrations, indicating the presence of indoor chemical sinks. In the case of ozone particularly, the personal indoor exposures were up to 25 times lower than the ambient concentrations, due to the high indoor reactivity of the pollutant. "

Comment 5: The application of personal measurements is increasing in the epidemiological studies, it's good to add some reviews on the progress in this area to highlight the advantage of this design.

95 Thank you for your suggestions. We have revised in the discussion section regarding the advantage of using PAM along with time activity model to exhibit the difference between ambient and personal exposure.

"Firstly, the study deployed a state-of-the-art and validated personal air pollution monitor to improve the personal exposure assessment to multiple pollutants. The high compliance rate of the participants with the study protocol highlighted the

feasibility of collecting personal exposure data at high spatiotemporal resolution matched with detailed health assessments.
The preliminary results highlight a clear difference between personal and ambient exposure driven by individual activity patterns, meteorological factors and the built environment. In line with previous literature, we show the large biases arising from the use of ambient measurements to represent personal exposure in most epidemiological studies, and the potential of novel sensor technologies to revolutionise future human-based studies.

Secondly, time-activity-location patterns of individuals are important determinants of personal exposure but due to the relative difficulty of collecting such information, they have rarely been taken into account by air pollution epidemiology. For the relatively sedentary participants of this panel study, the home environment was the major contributor to overall exposure, and an important modifier of personal concentrations for all investigated air pollutant species. Exposure differences between the two panels were attributed partly to the variation in domestic energy use. For instance, in winter the urban building stock in China relies on centralised gas heating system, while traditional biomass and coal stoves remain the key emission source for

110 heating and cooking in peri-urban areas. However, the exposure variability between participants was larger than the variability between the two groups, stressing the need to go beyond current methodologies to estimate population exposures."

**Response to referee #2**

This is an introduction paper on the protocol of the AIRLESS study conducted in Beijing. The overall study design is

- 115 rigorous in terms of the methods presented in exposure and health outcome measurements. The comparison of the health effects of air pollution in hypertensive population between urban and suburban area is innovative and interesting. The scope to explore comprehensive range of exposure and health outcome metrics is the strength of the study. Despite of this, there are several places in the manuscript that need to be improved by better clarifications of the methodology and preliminary results.
- 120 Overall: Thank you for taking the time to provide valuable feedback. This paper presented the methodological framework for the collection of detailed medical biomarkers and exposure estimates as part of the AIRLESS project. The comments are extremely relevant as the reviewer has a clear understanding of current gaps on the effects of air pollution on health and the potential of projects like AIRLESS to address such issues.

Regarding the reviewer's comment, we have added more tables and figures into the results and discussion session. The

125 manuscript has also been restructured and certain sections has been revised accordingly. Please find below the point-to-point response.

Comment 1: There are many air pollutants and health outcome measurements presented in this study. For air pollutants, it is possible that many species may come from the same sources and therefore, highly correlated. The association, if identified, may not directly reflect the true toxicity of health effect for a pollutant but an alternation of source-related effect. It is important to think more in the paper about how to make use of the comprehensive exposure data and propose novel method that can leverage the combination of multiple pollutants' effect in the health analysis. For the health outcomes, the multiple biomarkers from the same pathways may generate an issue of multiple comparison (or not). As this is a methodology paper, it would be helpful to include a discussion on this issue.

135 This comment hits the nail on the head. We understand it's really hard to differentiate the health impacts of species highly correlated in the outdoor environment due to similar sources (i.e. NO2 and PM2.5 both primarily emitted from traffic). The application of personal monitor and the developed time-activity-location model would be helpful to separate the effect of the key pollutant.

We have re-written parts throughout the manuscript to stress this point. We have modified the background section to stress the

- 140 four wider research gaps this project aims to address:
  - a) To investigate the interactive effects of air pollution and hypertension
  - b) To establish more reliable links between air pollution and health effects by reducing exposure misclassification.
  - c) To differentiate source-related health effects of air pollution
  - d) To investigate the underlying mechanism of air pollution on health

145

We also revised the manuscript in the discussion section to highlight the advantage of the use of personal exposure to find the difference between ambient personal exposure, and the application of time-activity model to potentially separate the health effect of highly correlated pollutant.

"Firstly, the study deployed a state-of-the-art and validated personal air pollution monitor to improve the personal exposure
assessment to multiple pollutants. The high compliance rate of the participants with the study protocol highlighted the feasibility of collecting personal exposure data at high spatiotemporal resolution matched with detailed health assessments. The preliminary results highlight a clear difference between personal and ambient exposure driven by individual activity patterns, meteorological factors and the built environment. In line with previous literature, we show the large biases arising from the use of ambient measurements to represent personal exposure in most epidemiological studies, and the potential of novel sensor technologies to revolutionise future human-based studies.

- Secondly, time-activity-location patterns of individuals are important determinants of personal exposure but due to the relative difficulty of collecting such information, they have rarely been taken into account by air pollution epidemiology. For the relatively sedentary participants of this panel study, the home environment was the major contributor to overall exposure, and an important modifier of personal concentrations for all investigated air pollutant species. Exposure differences between the
- 160 two panels were attributed partly to the variation in domestic energy use. For instance, in winter the urban building stock in China relies on centralised gas heating system, while traditional biomass and coal stoves remain the key emission source for heating and cooking in peri-urban areas. However, the exposure variability between participants was larger than the variability between the two groups, stressing the need to go beyond current methodologies to estimate population exposures." Regarding the examination of health outcomes, we reckon that an increasing number of biomarkers in a study will increase
- 165 the difficulties in explaining the biological mechanisms, as some of the biomarkers may share similar pathways or be regulated in a more complicated biological network (eg. Cytokines). The fast development of omic-related analysis, which could generate thousands of biomarkers, will be helpful but meanwhile add more challenge in understanding the biological mechanism. We have considered this issue for the analysis of multiple biomarkers, and revised the strategy for analysis accordingly in 2.9 statistical analysis, as followed:
- 170 "To examine the effect of air pollutant on multiple biomarkers (e.g. metabolome and transcriptome), false discovery rate (FDR) adjusted p-value will be applied to detect the statistical significance. Pathway enrichment analyses based on the changes in multiple biomarkers will be used to investigate potential mechanisms."

Comment 2: regarding to low-cost sensor technologies, "sensor technology is complex and requires careful calibration both internally within device and externally across the devices and with other standard instruments under various

175 environmental circumstances. The current study only reported the specifications and performances of the PAM monitor, but did not include detailed descriptions on how to ensure the accuracy of the monitors in the real world measurements."

Thanks for the comment from the reviewer. We understand the importance of the validation of the personal monitor (PAM) we used in this study. The performance of the sensors integrated in the PAM has been characterised extensively in a previous

180 publication (Chatzidiakou et al., 2019). We added a paragraph in the subsection of method (2.5 Personal exposure in revised manuscript) to summarize the key results from that paper, detailed as below: "The characterisation of the performance of the air quality sensors integrated in the PAM is presented in a previous publication (Chatzidiakou et al., 2019). Briefly, all PAMs were calibrated in two outdoor co-location deployments at the urban PKU site next to reference instrumentation for one month after the winter and summer deployments to participants. The performance of

- 185 the NO2 and PM2.5 sensors was additionally characterised in an indoor microenvironment next to commercial instruments. Overall, the air pollution sensors showed high reproducibility (mean R2= 0.93, min-max: 0.80–1.00) and excellent agreement with standard instrumentation (R2>0.84 for all sensors in winter, while R2>0.71 in summer). Further work (Chatzidiakou et al., 2020) showed that the error of the PAM was negligible compared with the error introduced when deriving exposure metrics from fixed ambient monitoring stations close to the participants' residential addresses. Hence, novel sensing technologies such
- 190 as the ones used here are suitable for collecting highly resolved personal exposure measurements in large-scale health studies." Comment 3: The results are relatively simple. At least, the demographics of the study population and the exposure and outcome measurement statistics are needed, so that it is good for readers to understand the overall differences of exposure and outcomes between the two study sites. The results will also help support the proposed hypothesis of the study.
- 195 Thanks for the comment. To better characterise the two panels of participants in this study, we have added a table (Table 4 in revised version). Basic health outcomes (such as BMI, WHR, hypertension status) were also included in Table 4. For the other detailed biomarkers, we plan to included in the following papers. Regarding to the exposure in the two sites, we added two figures (Figure 6 and 9 in revised version) to show the ambient and personal exposures during the campaigns in both sites.

We also revised the manuscript regarding to the results of these analysis, which is described in subsection 3.1 "Demographics 200 characteristics of urban and peri-urban participants", 3.2 "Ambient concentration of PM2.5 during study periods", and 3.5 "Seasonal and spatial pattern of the difference between personal and ambient exposure".

Comment 4: As mentioned by the authors, one of the major differences in the urban and suburban sites is the contribution of indoor exposure to the personal exposure and the indoor exposure levels are supposed to be higher. Only the outdoor monitoring sites include detailed air pollutant species as compared to the personal exposure. Thus, the immentance of the contributions of the specific to the personal exposure for the personal exposure.

205 the importance of the contributions of the species to the personal exposure seems to be attenuated. Will it be possible to use the GPS data to split outdoor and indoor exposure in the health analysis, so that the comparisons are relatively fair?

Thanks for the comment, this is really an important question to understand the health effect of pollutant species. Currently, personal sensors were not applicable to a wide range of pollutant species, especially considering the performance of

- 210 measurement at a high time-resolution. The common commercial portable monitors used in most of the epidemiological studies are usually targeted on either particles (PM2.5, BC, etc), or gaseous pollutants (NO2, CO etc). The personal monitor we used was developed ourselves which is unique to include both PM and gaseous sensors in one device. This enables the PAM to measure PM in different size fractions, and four species of gaseous pollutants, which will help us to understand the health effect of the key pollutants. We have updated Table 2 (combined the previous Table 2 and 3 together) in the manuscript to
- 215 describe the physical and chemical parameters of both ambient and personal exposure measurement.

Apart from that, we have also considered the suggestion of the reviewer to split indoor/outdoor exposure to give a more accurate exposure assessment. An automated model was developed to classify time-activity-location patterns based on parameters collected with the PAMs (GPS, background noise, acceleration)2, which has been described in our newly published paper. The classifications include core location categories: "home", "work", "other indoor static", "other outdoor static",

220 "travel", as well as activities "cooking", "sleeping" and modes of transport ("walk", "cycle", "motorbike", "car/bus", "train/tube").

New subsection (2.6 The time-activity-location model) was added to highlight this methodological element of the project. We also added a figure (Figure 8 in revised manuscript) and a subsection in result (3.4 an illustrative example of exposure misclassification) to show how we apply the time-activity-location to help understanding the potential sources for personal

**exposure. References:**

1 Chatzidiakou, L., Krause, A., Popoola, O. A., Di Antonio, A., Kellaway, M., Han, Y., ... & Fan, Y. (2019). Characterising low-cost sensors in highly portable platforms to quantify personal exposure in diverse environments. *Atmospheric measurement techniques*, *12*(8), 4643.

230

**List of changes in the manuscript**

**Tables and Figures**

- 1. Merged Table 2 and 3 into one table (Table 2 in the revised manuscript)
- 2. Change Table 4 as Table 3 in the revised manuscript (Table 3: Measurement plans for health outcomes in AIRLESS study)
- Move Table 5 to supplement (Table S1: The minimum detectable effects for the 4 cardiopulmonary outcomes in crosssectional and longitudinal settings)
  - 4. Summarize characteristics of the participant and showed in a new table (Table 4 in the revised manuscript)
  - 5. Remove Table 6 as the information of hypertension is covered in the updated Table 4
  - 6. Move Figure 4 to supplement (Figure S1: Personal Air Monitor (PAM) used in AIRLESS Model and Applications)
- 240 7. Add a new figure (Fig 6 in the revised manuscript) to show the ambient PM2.5 concentrations in both sites during the monitoring campaigns
  - 8. Add a new figure (Fig 7 in the revised manuscript) to show the completeness of personal exposure measurement
  - 9. Add a new figure (Fig 8 in the revised manuscript), using a certain participant as an example to show the difference between personal and ambient exposure
- 245 10. Add a new figure (Fig 9 in the revised manuscript) to show the Seasonal and spatial pattern of the difference between personal and ambient exposure

**Manuscript**

- 1. Modified the background section to stress the four wider research gaps this project aims to address:
  - a) To investigate the interactive effects of air pollution and hypertension
- b) To establish more reliable links between air pollution and health effects by reducing exposure misclassification.
  - c) To differentiate source-related health effects of air pollution
  - d) To investigate the underlying mechanism of air pollution on health
  - 2. Methods (2.1 Aims and objectives), highlight the time-activity-location model.
  - 3. Add a paragraph to describe the validation of PAM (subsection 2.5 "Personal Exposure" in the revised manuscript).
- Add a paragraph to describe the time-activity-location model and the application in this study (subsection 2.6 "he timeactivity-location model" in the revised manuscript).
  - 5. Restructure orders/numbers of subsections in Methods.
  - Add a subsection to describe the characteristics of participants in two sites (subsection 3.1, "Demographics characteristics of urban and peri-urban participants").
- Add a subsection to describe the ambient PM2.5 concentration in two sites across the campaigns (subsection 3.2, "Ambient concentration of PM2.5 during study periods").
  - Add a subsection to describe the completion of personal exposure measuremnets (subsection 3.3, "Completion of personal exposure during study periods").

9. Add a subsection to illustrate an example of the difference between personal and ambient exposure (subsection 3.4, "An

- illustrative example of exposure misclassification").
  - 10. Add a subsection to show the preliminary results of the comparison between personal and ambient exposure in all the participants (subsection 3.5, "Seasonal and spatial pattern of the difference between personal and ambient exposure").
  - 11. Revise the discussion section to highlight the strength of the study (use of personal monitor, application of time-activitylocation model, and collection of rich dataset in exposure and health outcomes).
- 270 12. The manuscript has also been revised for a clear expression.

[revised manuscript text omitted]
 qua | lity and pub   | health in the la     | st three decades. L  | riven by the  |
|-----|--------------------------|--------------------|---------------|-----------------|----------------|----------------------|----------------------|---------------|
| 675 | Givencussion the         | severe             | air p         | ollution        | and            | nationwide           | hypertension         | epidemic      |
|     | in China, AIRLESS s      | ets to (a) investi | gate the inte | ractive effects | of air polluti | on and hypertens     | ion, (b) establish n | nore reliable |
|     | links between air poll   | lution and health  | effects by    | reducing expos  | sure misclass  | sification, (c) diff | erentiate source-re  | elated health |
| 680 | effects of air pollution | n, and (d) investi | gate the und  | erlying mechar  | nism of air po | ollution on health.  | Several novel me     | thodological  |
|     | elements str             | rengthen           | the           | design          | of             | the                  | AIRLESS              | study.        |

685

Firstly, the study deployed a state-of-the-art and validated PAM to improve the personal exposure assessment to multiple pollutants. The high compliance rate of the participants with the study protocol highlighted the feasibility of collecting personal exposure data at high spatiotemporal resolution matched with detailed health assessments. The preliminary results highlight a clear difference between personal and ambient exposure driven by individual activity patterns, meteorological factors and the built environment. In line with previous literature, we show the large biases arising from the use of ambient measurements to

represent personal exposure in most epidemiological studies, and the potential of novel sensor technologies to revolutionise future human-based studies.

Secondly, time-activity-location patterns of individuals are important determinants of personal exposure but due to the relative difficulty of collecting such information, they have rarely been taken into account by air pollution epidemiology. For the

- 695 relatively sedentary participants of this panel study, the home environment was the major contributor to overall exposure, and an important modifier of personal concentrations for all investigated air pollutant species. Exposure differences between the two panels were attributed partly to the variation in domestic energy use. For instance, in winter the urban building stock in China relies on centralised gas heating system, while traditional biomass and coal stoves remain the key emission source for heating and cooking in peri-urban areas. However, the exposure variability between participants was larger than the variability
- 700 between the two groups, stressing the need to go beyond current methodologies to estimate population exposures. Last, panel studies might be the most suitable way to link intensive air monitoring campaigns for a wide range of pollutant species, personal exposure in different micro-environments, and together with epidemiological studies of detailed biological changes in human. Taking advantage of the simultaneously launched air monitoring campaigns we successfully collected a rich set of data regarding both exposure and health outcomes. This provides a rare opportunity to investigate the effect of

705 different pollutant species and the underlying biological pathways.

Altogether, the forthcoming outcomes of the AIRLESS project will enhance our understanding of the impact of environmental Altogether, the forthcoming outcomes of the AIRLESS project will enhance our understanding of the impact of environmental Altogether, the forthcoming outcomes of the AIRLESS project will enhance our understanding of the impact of environmental Altogether, the forthcoming outcomes of the AIRLESS project will enhance our understanding of the impact of environmental Altogether, the forthcoming outcomes of the AIRLESS project will enhance our understanding of the impact of environmental Altogether, the forthcoming outcomes of the AIRLESS project will enhance our understanding of the impact of environmental Altogether, the forthcoming outcomes of the AIRLESS project will enhance our understanding of the impact of environmental Altogether, the forthcoming outcomes of the AIRLESS project will enhance our understanding of the impact of environmental Altogether, the forthcoming outcomes of the AIRLESS project will enhance our understanding of the impact of environmental Altogether, the forthcoming outcomes of the AIRLESS project will enhance our understanding of the impact of environmental Altogether, the forthcoming outcomes of the AIRLESS project will enhance our understanding of the impact of environmental Altogether, the forthcoming outcomes of the AIRLESS project will enhance our understanding of the impact of environmental Altogether, the forthcoming outcomes of the AIRLESS project will enhance our understanding of the impact of environmental Altogether, the forthcoming outcomes of the AIRLESS project will enhance our understanding of the impact of environmental Altogether, the forthcoming outcomes of the AIRLESS project will enhance our understanding of the impact of environmental Altogether outcomes of the AIRLESS project will enhance outcomes of the AIRLESS p

710 Altogether, the forthcoming outcomes of the AIRLESS project will enhance our understanding of the impact of environmental Altogether, the forthcoming outcomes of the AIRLESS project will enhance our understanding of the impact of environmental Altogether, the forthcoming outcomes of the AIRLESS project will enhance our understanding of the impact of environmental

**Abbreviations**

[revised manuscript text omitted]

Tables

**Table 1: Recruitment Criteria**

| Inc          | lusion                    |  |
|--------------|---------------------------|--|
| $\checkmark$ | $50 \le age \le 75$ years |  |

- > Non-smoker or those who has quitted smoking longer than 3 years
- Hypertensive subject participants (clinical diagnosis\*)
- Healthy subjectparticipants (clinical diagnosis\*)

**Exclusion**

Diagnosed with disease history of any cardiovascular or metabolic diseases, including Hyperlipidemia, malignant tumour, Coronary Heart Disease, Cardiomyopathy, Arrhythmia, Stroke, Hepatitis A/B, Leukaemia, Biliary calculus, Thyroid nodule, sick sinus syndrome or wearing cardiac pacemaker, Hyperthyroidism, Hypothyroidism, Multiple myeloma, Rheumatoid arthritis, Pancreatitis, Reflux esophagitis, Thyroidectomy

\* Clinical diagnosis: systolic blood pressure (SBP) >= 140 mmHg or diastolic blood pressure (DBP) >= 90 mmHg occurred 915 in two repeated measurements

| Exposure Index                             | Parameters                                                                            | Method/Instrument                           | Resolution 920 |
|--------------------------------------------|---------------------------------------------------------------------------------------|---------------------------------------------|----------------|
|                                            | Ambient E                                                                      | xposure                                     |                |
|                                            | PM 2.5 mass concentration                                                  | BAM(Pinggu) / TEOM 1400a (PKU)              | Hourly         |
|                                            | BC mass concentration                                                                 | MAAP (Pinggu)/AE33 (PKU)                    | Hourly         |
|                                            | Size distribution                                                                     | SMPS                                 | Hourly  |
|                                            | Online EC/OC                                                                          | Sunset                                      | Hourly  |
| Particulate
Pollutants                  | Metal Element                                                                         | Xact                                        | Hourly         |
| r onutants                          | NR-Chemical Composition                                                               | ACSM/TOF-ACSM                               | Hourly  |
|                                            | Water Soluble Ions
Water Soluble Organic Acid                                      | Dionex ICS-2500/2000
Liquid Chromatogram | Daily   |
|                                            | Metal Element                                                                         | Thermo X series ICP-MS                      | Daily   |
|                                            | PAHs                                                                                  | Agilent GC-MS                               | Daily   |
|                                            | CO                                                                             | NDIR/Thermo Model 48i                       | Minute         |
| Gaseous Pollutants                         | NOx                                                                                   | Chemiluminescence/
Thermo Model 42i      | Minute         |
| ouseous romanants                   | SO2                                                                            | Fluorescence/Thermo Model 43c               | Minute         |
|                                            | O 3                                                                 | UV absorption/ Thermo Model 49i             | Minute         |
| Meteorological
Parameters | Temperature, relative humidity,
Barometric pressure, wind speed,
wind direction | Met One                                     | Minute         |
|                                            | Personal E                                                                     | xposure                                     |                |
| Particulate
Pollutants    | PM1, PM2.5, PM10 mass
concentration                                         | Optical Particle Counter (OPC)              | 20 sec  |
| Gaseous Pollutants                         | CO, NO, NO2, O3                                          | Electrochemical sensors                     | 20 sec  |
| Meteorological                             | Temperature                                                                           | thermocouple                                | 1 min   |
| Parameters                                 | Relative Humidity (RH) (%)                                                            | Electrical resistive sensor                 | 1 min   |
|                                            | Spatial coordinates                                                                   | Global Positioning System (GPS)             | 1 min   |
| Activity                                   | Background noise                                                                      | Microphone                                  | 100 Hz  |
|                                            | Physical activity                                                                     | Tri-axial accelerometer                     | 100 Hz  |

**Table 2: The matrix of exposure parameters in the AIRLESS study**

| Biological Pathways                                                           | Sample/Device                                              | Health Endpoints                                                                                                                   |                              |
|--------------------------------------------------------------------------------------|------------------------------------------------------------|------------------------------------------------------------------------------------------------------------------------------------|------------------------------|
| Blood Pressure and Heart Rate                                                 | Omron HEM 907                                              | Systolic Pressure/Diastolic Pressure/Heart Rate                                                                                    |                              |
| Endothelial Function                                                                 | Pulse wave analyzer                                        | AP/ <del>AP75/</del> AIx/ <del>AIx75/</del> ED/SEVR                                                                                |                              |
|                                                                                      | Peak flow metre                                            | PEF                                                                                                                                |                              |
| Doonington Inflommation                                                              | Exhaled Breath                                             | FE NO                                                                                                                   |                              |
| Respiratory mnammation                                                               | EDC                                                        | pH                                                                                                                                 |                              |
|                                                                                      | EDC                                                        | Cytokines: e.g. IL1α/IL1β/IL2/IL6/IL8/TNF-α/IFN-γ                                                                                  |                              |
|                                                                                      | Samum                                                      | CRP                                                                                                                                |                              |
| Cardiovascular Inflammation                                                          | Serum                                                      | Cytokines: e.g. IL1α/IL1β/IL2/IL6/IL8/TNF-α/IFN-γ                                                                                  |                              |
|                                                                                      | Plasma                                                     | WBC/neutrophil/monocyte/lymphocyte                                                                                                 |                              |
| Matabalia                                                                            | S                                                          | TG/HDL/LDL/cholesterol                                                                                                             |                              |
| Metabolic                                                                            | Serum                                                      | Glucose/Insulin/HOMA-IR                                                                                                            |                              |
|                                                                                      | Serum/Urine                                                | Untargeted/targeted Metabolomic signatures                                                                                         |                              |
| 0.11.12 04                                                                           | Urine                                                      | MDA/Creatinine                                                                                                                     |                              |
| Oxidative Stress                                                                     | Plasma                                                     | DNA repair enzyme                                                                                                                  |                              |
| Genetic related pathways                                                             | Blood                                                      | Genetic and Epigenomic profiles                                                                                                    |                              |
| P: augmentation pressure; AIx:                                                       | augmentation index; E                                      | BC: Exhaled breath condensate; PEF: Peak expiratory flow;                                                                          | Formatted: Left, Indent: Fin |
| ENO: Fractional exhaled NO; IL
lood cells: TG: Triglvceride: HE | : Interleukin; IFN-γ: In
DL: High–density lipopr | terferon- $\gamma$ ; TNF- $\alpha$ : tumor necrosis factor- $\alpha$ ; WBC: white
otein: LDL: Low-density lipoprotein: HOMA-IR: | Formatted: Subscript         |
| Iomeostatic model assessment of                                                      | insulin resistance; MD                                     | A: Malondialdehyde; ED: Ejection duration; SEVR:                                                                                   |                              |
| ubendocardial viability ratio                                                        |                                                            |                                                                                                                                    | Formatted: Font: (Asian) +H  |

**Table 3: Measurement plans for health outcomes in AIRLESS study**

930

925

| Table 4: Statistic summar                                     | ry of demographic charact | teristics of urban and  | peri-urban particip | ants            |
|---------------------------------------------------------------|---------------------------|-------------------------|---------------------|-----------------|
| A                                                      | Unit                      | Urban                   | Peri-urban          | P value*        |
| Participant (winter)                                          | N                  | 123              | 128          |                 |
| Participant (summer)                                          | N                  | 102                     | 116          |                 |
| Visit person-times                                            |                           |                         |                     |                 |
| All                                                           | N                  | 450              | 488                 |                 |
| Winter                                                        | N                         | 246              | 256          |                 |
| Summer                                                 | N                  | 204              | 232          |                 |
| Participants Statistics
Continuous Variables | Mea                       | n (standard deviation   | 1, SD)       |                 |
| Age                                                           | Years                     | 65.7 (4.4)              | 60.7 (5.5)          | < 0.01          |
| BMI                                                           | kg/m 2         | 24.8 (3.2)              | 26.4 (3.2)          | <0.01 |
| WHR                                                           | NA                        | 0.87 (0.05)      | 0.89 (0.04)  | <0.01 |
| Participants Statistics                                       | N (per                    | contage of total partic | cinante)            |                 |
| Categorical Variables                                         |                           | centage of total partic | cipants)     |                 |
| Gender                                                 |                           |                         |                     |                 |
| Male                                                          | #(%)               | 58 (47.2)        | 51 (39.8)    | 0.26            |
| Female                                                 | #(%)               | 65 (52.8)               | 77 (60.2)           |                 |
| Education                                                     |                           |                         |                     |                 |
| High school and below                                         | #(%)               | 27 (22.0)               | 128 (100.0)         | < 0.01          |
| College and above                                             | #(%)               | 96 (78.0)        | 0 (0.0)      |                 |
| Annual Income                                                 |                           |                         |                     |                 |
| <20,000 RMB                                         | #(%)               | 8 (6.5)          | 67 (52.3)    |                 |
| ≥20,000 RMB                                            | #(%)               | 111 (90.2)       | 53 (41.4)    | <0.01 |
| NA                                                     | #(%)               | 4 (3.3)          | 8 (6.2)             |                 |
| Smoking Status                                                |                           |                         |                     |                 |
| Non-smoker                                                    | #(%)               | 99 (80.5)               | 99 (77.3)           | 0.62            |
| Past-smoker                                                   | #(%)               | 24 (19.5)               | 29 (22.7)    | 0.03            |
| Secondhand Smoking*                                           |                           |                         |                     |                 |
| Never                                                         | #(%)               | 73 (59.3%)       | 65 (50.8%)   |                 |
| Past                                                          | #(%)               | 30 (24.4%)              | 26 (20.3%)          | <0.05           |
| Now                                                           | #(%)               | 19 (15.4%)       | 37 (28.9%)          | <0.05 |
| NA                                                     | #(%)               | 1 (0.8%)         | 0 (0%)       |                 |
| Cooking Time                                                  |                           |                         |                     |                 |
| <1h/day                                             | #(%)               | 64 (52.0%)              | 48 (37.5%)          |                 |
| >=1h/day                                                      | #(%)               | 57 (46.3%)       | 79 (61.7%)   | <0.05 |
| NA                                                            | #(%)               | 2 (1.6%)         | 1 (0.8%)     |                 |
| Hypertension                                                  |                           |                         |                     |                 |
| No                                                            | #(%)               | 66 (53.7%)       | 81 (63.3%)   | 0.16            |
| Yes                                                           | #(%)               | 57 (46.3%)       | 47 (36.7%)          |                 |
| Hypertension Medication                                       | 11/0/ >                   | 72 (50 20()             | 00 (60 50)          | 0.52            |
| No                                                     | #(%)               | 73 (59.3%)              | 80 (62.5%)          | 0.53            |

Table 4: Statistic summary of demographic characteristics of urban and peri-urban participants

| Yes              | #( %)                                 | 48 (39.0%)        | 43 (33.6%)            |          |
|------------------|----------------------------------------------|--------------------------|-----------------------|----------|
| NA               | #(%)                                  | 2 (1.6%)          | 5 (3.9%)       |          |
| *The significant | nee of difference between the urban and pari | urbon participants and t | ha n valua is datarmi | nod boso |

\*The significance of difference between the urban and peri-urban participants and the p-value is determined based on student t test and chi-square test for continuous and categorical variables, respectively

**Figures**